# Analysis of the Suitability of Ultrasonic Testing for Verification of Nonuniform Welded Joints of Austenitic–Ferritic Sheets

**DOI:** 10.3390/ma17174216

**Published:** 2024-08-26

**Authors:** Łukasz Rawicki, Ryszard Krawczyk, Jacek Słania, Grzegorz Peruń, Grzegorz Golański, Katarzyna Łuczak

**Affiliations:** 1GIT Łukasiewicz Research Network—Upper Silesian Institute of Technology, K. Miarki 12-14, 44-100 Gliwice, Poland; lukasz.rawicki@git.lukasiewicz.gov.pl (Ł.R.); jacek.slania@git.lukasiewicz.gov.pl (J.S.); 2Faculty of Mechanical Engineering and Computer Science, Częstochowa University of Technology, Armii Krajowej 21, 42-201 Częstochowa, Poland; ryszard_krawczyk@wp.pl; 3Department of Road Transport, Faculty of Transport and Aviation Engineering, Silesian University of Technology, Krasińskiego 8, 40-019 Katowice, Poland; 4Department of Material Engineering, Częstochowa University of Technology, Armii Krajowej 19, 42-201 Częstochowa, Poland; grzegorz.golanski@pcz.pl; 5Faculty of Architecture, Civil Engineering and Applied Arts, Academy of Silesia, Rolna 43, 40-555 Katowice, Poland; katarzyna.luczak@wst.com.pl

**Keywords:** ultrasonic inspection, austenitic steel, ferritic steel, DAC, 13CrMo4-5, X2CrNiMo17-12-2 (AISI 316L)

## Abstract

The purpose of the presented research was to determine the suitability of using ultrasonic testing (UT) to inspect heterogeneous, from a material point of view, welded joints on the example of the joints of a ferritic steel element with elements made of an austenitic steel. The echo technique with transverse (SEK) and longitudinal wave heads (SEL) addressed this issue. Due to the widespread use of 13CrMo4-5 and X2CrNiMo17-12-2 steel grades in the energy industry, they were selected as the test materials for the study. The objects of the presented research were welded joint specimens with thicknesses of 8, 12, and 16 mm and dimensions of 300 × 300 mm, made using the 135 metal active gas (MAG) process with the use of the Lincoln 309LSi wire—a ferritic–austenitic filler material. The stages of the research task were (1) making distance–amplitude curve (DAC) patterns from the test materials; (2) preparation of specimens of welded joints with artificial discontinuities in the form of through-holes; (3) performing UT tests on welded joints with artificial discontinuities using heads with 60° and 70° angles for the transverse wave and angle heads for longitudinal waves with similar beam insertion angles; (4) selection, by radiographic testing (RT), of welded joint specimens with natural discontinuities in the form of a lack of sidewall fusion; (5) performing UT tests on welded joints with natural discontinuities, using heads as welded joints with artificial discontinuities. It was found that (1) the highest sensitivity of discontinuity detection was obtained by performing tests on the ferritic steel side, which is due to the lower attenuation of the ultrasonic wave propagating in ferritic steel compared to austenitic steel; (2) the best detection of discontinuities could be obtained using a longitudinal ultrasonic wave; (3) there is a relationship between the thickness of the welded elements, the angle of the ultrasonic beam introduction, and the effectiveness of discontinuity detection.

## 1. Introduction

Welding processes are classified as special processes, that is, processes whose results cannot be predicted while they are in progress and potential imperfections may only become apparent after completion. It is therefore crucial to test welded joints, with the recommended methods being volumetric non-destructive methods, which include ultrasonic testing (UT). In the case of heterogeneous joints, the issue with carrying out ultrasonic testing is the differing physical properties of the various zones of the welded joint, which are related to the conditions under which these tests are performed. For example, in UT testing of austenitic steels, the problem is wave transformation, limited detection of subsurface weld discontinuities, and abnormal coupling due to surface roughness. These issues are discussed in more detail in [1,2]. The coarse-grained structure causes scattering and attenuation of the wave and changes its direction. Attenuation affects the pressure amplitude of the wave propagating through the material, which decreases with increasing distance from the head [3]. In the case of heterogeneous joints, there are differences in the measured amplitude values in different areas of the joint. The primary factors limiting the detection of discontinuities are beam scattering and divergence, which are influenced by the material’s structural properties. These phenomena are particularly pronounced in the assessment of nonuniform welded joints [4].

The necessity to join materials with significantly disparate physical properties by welding methods is a common occurrence in the field of mechanical engineering, including in the construction of power equipment, reactors, and chemical installations. For instance, in power boilers, heat exchanger tubes operating at exceedingly high temperatures are manufactured from austenitic steels and joined with plant components fabricated from ferritic steels. Austenitic–ferritic and duplex steels are employed in the fabrication of chemical tankers. Furthermore, structural elements in joints with tank components made of duplex steels are also examples of joints of heterogeneous materials, in this case usually with high-strength carbon steel [5]. The popularity of these materials in the construction of nuclear power plant installations around the world is due to their high durability and corrosion resistance. In this case, safety considerations require the implementation of tests to determine the correctness of the technical condition of the components, which are carried out by non-destructive testing at specified intervals. One of the main methods of implementing such tests, despite the difficulties outlined earlier, is the ultrasonic technique.

The fundamental principle of ultrasonic testing, assuming a known and constant speed of sound, is to relate the time of the received echo to the location of the reflector. However, in the case of austenitic welded joints, where the microstructure with large oriented austenitic grains induces local velocity differences that deflect the ultrasonic beam, this assumption is not valid [6]. Furthermore, the testing problem is complicated by scattering at grain boundaries, causing structural noise and attenuation. For this reason, a number of solutions have been proposed, including the embedding of material information and the weld structure into imaging algorithms. This usually improves the quality of the images and aids interpretation [7].

One of the research projects at UT that was engaged in the study of the nuclear industry is [8]. The study examined the ultrasonic behavior of beams in relation to the orientation of grains in the weld joint. The study employed both conventional ultrasonic transducers and linear and matrix phase transducers to generate ultrasonic beams. To enhance the efficacy of ultrasonic non-destructive testing within this industry, a methodology for modeling ultrasonic inspection of such welds, which has been successfully applied to arc welding, was presented in Ref. [9]. The paper also presented the results of laboratory tests on the repair of a nickel-based alloy weld made using an automatic tungsten arc welding process, with particular attention paid to ultrasonic propagation interferences, including anisotropic, heterogeneous, and coarse-grained weld structures of austenitic stainless steel.

The primary factors contributing to the difficulty of inspecting austenitic steels and joints of these steels with ferritic steels by degradation of image quality presented in Ref. [10] are time errors, phase errors, and multipath propagation and scattering. They were selected based on experimental studies conducted during the inspection of austenitic welds with high inhomogeneity and anisotropy.

During ultrasonic testing of nonuniform welded joints, measurement parameters are very often chosen experimentally. This is due to the fact that, at present, no standards and regulations specify criteria for evaluating volumetric testing of such joints by ultrasonic methods. This makes it necessary to determine test parameters that will provide the necessary conditions to detect discontinuities in the analyzed case of welded joints PN-EN ISO 11666:2018-04 [11].

The purpose of this paper is to present the methodology and to learn the cognitive qualities that are the starting point for the study of welded joints of miscellaneous materials. Due to their peculiarities, joints of this type require a different approach to the study of classical ferritic steels. The testing capabilities have been verified on reference specimens made from the test materials. This type of joint is used in the energy, petrochemical, and chemical industries. A heterogeneous welded joint is, among other things, a configuration of two different base materials and the welding of base materials with an additional material with different properties.

## 2. A Brief Overview of UT Research Directions in the Literature

The testing of ferritic steels with UT generally poses no difficulties. The most common technique for testing ferritic welds is the echo method, which utilizes vibration frequencies of 2–5 MHz in materials with a fine-grained structure. However, the issue of testing austenitic steels is a significant concern for many researchers, with several works, including [12,13], providing detailed descriptions of the problem. The large grain size and high ultrasonic attenuation are identified as the source of the problem. The authors note the difficulty of penetrating deep into the material, since stainless steel is also an anisotropic material, meaning that ultrasonic properties vary depending on the direction of propagation. It has been demonstrated that conventional techniques have shown limited success in inspecting welded joints and that the use of various phase matrix techniques is recommended for the inspection of stainless steel components. Ref. [14] addresses the subject of offline post-processing techniques using the full set of time-domain data from all combinations of transmit and receive elements. The authors define the mathematical and practical implementation of four post-processing algorithms for use with the full matrix of array data: plane B-scan, focused B-scan, sector B-scan, and the total focusing method (TFM). The main advantages of these approaches are increased sensitivity to small defects and increased inspection coverage.

In Ref. [15], the application of mechanized ultrasonic inspection to the examination of austenitic stainless steel test blocks with various types of defects, including intergranular stress corrosion cracking, is described. The results demonstrated that cracks located in the weld’s heat-affected zone are relatively easily detected due to the ability to perform inspection from both sides of the weld. However, in the event of restricted accessibility, when the inspection is conducted solely from one side of the weld, the differentiation of signals emanating from the dissipation within the weld and those indicative of cracks presents a significant challenge.

The influence of alloy composition and welding line energy of two Ni-Cr-Fe alloys on the crystallographic structure and the characterization of the effect on ultrasonic propagation was investigated in Ref. [16]. Samples made entirely of weld metal were examined using ultrasonic testing, and then, characterized using large-area backscattered electron diffraction. The pseudo-single-crystal structure exhibited significant variations in the UT scan outcomes when the ultrasonic beam angle was altered in relation to the texture direction. However, the observed discrepancies in the UT scans with regard to the texture direction were less pronounced, attributable to the fact that the fiber texture was predominantly oriented along a single crystallographic axis and tended to be random in other orientations.

Ref. [17] presents a novel phase-matching technique, designated as “reverse phase matching”. The technique employs a ray-tracing algorithm to model the propagation of an acoustic wave and calculate the sound propagation time, taking into account the anisotropy and inhomogeneity of the weld structure. According to the researchers, this technique can be used to reconstruct 2D and 3D images in real time. An iterative gradient constant descent method (GECDM) algorithm was implemented, which is essential for studying inhomogeneous anisotropic media with unknown properties.

The difficulties and complications associated with the application of NDT techniques to the peripheral welds of a urea reactor are examined in detail in [18]. The authors conducted a comprehensive review of the theory and advanced aspects of UPA technology and demonstrated that UPA is effective and efficient in solving the safety inspection of circumferential welds. Subsequently, two related test blocks (RB-2/20 and an austenitic stainless steel butt joint test block) with artificial defects were selected to simulate the outer and inner layers.

Ref. [19] presents phased array ultrasonic testing (PAUT) of trimetallic welded joints in a prototype fast breeder reactor (PFBR) using dual matrix array (DMA) probes. The inspection of SS316LN–Alloy 800 welds is challenging due to the anisotropic and inhomogeneous properties of the weld pool, which leads to beam skewing and affects the detection, location, and size of discontinuities. The PAUT test was conducted on an SS316LN to Alloy 800 welded joint section of trimetallic welded joints and verified by RT. A weld specimen (24 mm thick) with an artificial defect such as lack of sidewall fusion (LOF) was produced by metal arc welding and gas tungsten arc welding. The study demonstrated that the DMA probe successfully detected LOF defects and improved the signal-to-noise ratio in the inspection of trimetallic welded joints. This shows that the DMA probe is a promising candidate for the inspection of coarse-grained, anisotropic, and heterogeneous materials.

A specific research objective is presented in Ref. [20]. The authors decided to utilize the laser ultrasonic technique to identify grain size in interstitial steels (IFs). They assert that the ultrasonic technique can be employed online for direct grain size measurements during steel production. The absolute values of the average grain size were calculated directly from the ultrasonic traces. In turn, in [21], UT tests were conducted in order to determine the orientation of the grains in an anisotropic weld. The objective was to facilitate future ultrasonic NDT simulations of these types of welds, with the aim of obtaining more accurate results. A model was introduced, assuming the type of weld, the transmitter model, the receiver model, and the type of two-dimensional ray tracing algorithm.

The coarse-grained, dendritic grain structure of the weld material and its anisotropy are features detailed in numerous works on ultrasonic testing of austenitic welds [22,23]. The influence of these factors on ultrasonic wave propagation and their distortions has led many researchers to conclude that the use of conventional ultrasonic research techniques using constant beam angles is very limited and the use of phased array ultrasonic techniques is desirable. The enhancement of ultrasonic signals for coarse-grained materials through machine learning analysis is described in [24].

Ultrasonic array technology offers significant advantages over conventional ultrasonic technology, mainly due to its ability to focus and direct the beam and its electronic scanning capabilities [25]. Despite its advancement, ultrasound examination using this method still provides results that are difficult to evaluate. In the case of cast stainless steels, UT does not permit the achievement of sufficient measurement accuracy. In order to address this issue, in [26] a comprehensively analyzable phased array ultrasonic testing system (PAUT) was developed. The authors noted that the defect dependence of echo intensity differs from granular noise when PAUT conditions, such as focusing depths and ultrasonic incidence angles, are continuously changed.

## 3. Testing Techniques for Welded Joints

The efficacy of ultrasonic testing in detecting discontinuities is contingent upon a number of variables, including the characteristics of the ultrasonic beam and the ability of the discontinuities to reflect the wave. As previously demonstrated in [27], if the size of the discontinuity is less than half the wavelength, it will be bypassed and the discontinuity will not be detected, with the ultrasonic wave undergoing only slight scattering. This is related to the roughness of the tested surface. High roughness, characterized by discontinuities, causes strong scattering of the beam. However, it should be noted that part of the ultrasonic beam may reach the head and be analyzed by the flaw detector system. Surfaces characterized by low roughness will reflect the ultrasonic beam well. When performing ultrasonic tests, the angle at which the ultrasonic wave beam is introduced into the tested material is also important. The largest echo can be obtained when the wave beam falls perpendicularly onto the surface of the discontinuity. This is in accordance with the findings presented in [28]. The angle formed by the direction of the incident wave and the plane of discontinuity is larger at higher frequencies than with head transducers using lower frequencies. At lower frequencies, the beam is more divergent, and part of the reflected wave more easily reaches the head at a specific location of discontinuities in the material.

The most frequently used technique for testing ferritic welds is the echo method. With a fine-grained material structure, vibration frequencies of 2–5 MHz are utilized. In an austenitic weld, a coarse-grained structure is formed as a result of the crystallization of the basic and additional materials. Consequently, a weld pool will be formed whose crystallite direction is parallel to the material thickness [29]. The side edges of the weld may be characterized by the deflection of crystallites in a direction perpendicular to the direction of the base material. In the case of austenitic welds, they are usually 10 mm long and 1 mm wide. The greater length may be the result of the greater thickness of the welding beads. The size of the grains and their exact orientation may depend on the type of material or technology used. The austenitic weld structure may exhibit features that impede the use of volumetric testing methods, such as the ultrasonic method. These features include the presence of elongated grain sizes, their orientation in the direction of thickness, or deviations in the orientation of crystallites near the fusion line.

The ultrasonic testing of various joints between low- and high-alloy steels and nickel alloys is typically more difficult than testing joints made of ferritic steels [30]. The issue of testing various welded joints is considered in terms of the structure influencing the propagation of the ultrasonic wave, the measurement technique, and the evaluation criteria. The factors that play an important role are the test parameters such as the ultrasonic wave used, the transmitting pulse, and the reduction in structural noise. The difficulties associated with conducting ultrasonic tests in welds of various joints may be caused by the coarse-grained or directionally oriented structure of the material, which can result in significant differences in attenuation, reflection, and refraction of waves at the grain boundaries. The lack of significant differences in the structure of the weld and the base material does not alter the procedure under the test conditions. The structure of the native material is usually fine-grained and does not pose any problems for the propagation of the ultrasonic beam. In such a case, the test is carried out similarly to the testing of ferritic steels. The structure of austenitic joints is most often heterogeneous and coarse-grained. The grains are oriented in the direction of heat dissipation, and the sizes of columnar crystallites often exceed several millimeters [31,32].

The structure of welds made of various materials gives rise to the occurrence of apparent indications which should not be taken into account when performing the test. In instances where indications are observed that do not originate from typical welding discontinuities, it has been found that the propagation speeds of ultrasonic waves are different. Furthermore, the acoustic wave resistances (impedance) are also different, and these values are characteristic of the type of wave and medium. High-alloy steels with a high chromium content, above 10%, may be prone to the segregation of components and excessive grain growth. The transition zone is particularly vulnerable to this phenomenon. Such phenomena may result in the appearance of false echoes that are difficult to interpret when performing ultrasound examinations [33].

Appropriate testing methodologies, taking into account sufficient sensitivity, enable the development of testing techniques for these welds. Oblique heads of transverse waves enable the examination of welds up to 25 mm thick. When examining welds whose thickness exceeds 25 mm, it is convenient to use oblique longitudinal wave heads. Testing with transverse waves polarized parallel to the surface where the waves are introduced also gives good results. The testing of welded joints of various materials should be based on the test results of test joints made using the same technological parameters. In this type of testing, an important issue is the possibility of access from both sides of the joint, knowledge of the grain orientation, and an appropriately processed face surface. In order to test such joints, a detailed test procedure must be developed. This procedure should include provisions related to testing standards and provide possible exceptions. The elimination of difficulties encountered when testing materials with a strongly damping structure can be carried out by using appropriate measurement parameters such as the selection of the ultrasonic wave, frequency, and head angle [34].

The study of various connections, including low-alloy and austenitic steel, encounters difficulties related to the coarse-grained structure. Another issue is anisotropy, which is related to the direction-dependent wave propagation speed [35,36]. When testing diverse joints, another problem is the transformation of the ultrasonic wave occurring at the fusion line between the base material and the austenitic weld. The propagation of the ultrasonic beam is not linear, rather it exhibits a bending behavior in the vicinity of fusion lines and grain boundaries. This phenomenon results in beam splitting and the generation of virtual echoes due to the geometry of the shape [37].

The structural construction of various welded joints differs. Given the absence of a uniform welded joint comprising multiple sheets of steel, it is essential to select a testing technique that considers the properties of the base material, weld material, thickness of the tested element, welding method, and parameters, in accordance with the standards set forth in PN-EN ISO 22825:2017-12 [38].

Heterogeneity and a coarse-grained anisotropic structure result in incorrect localization of indications detected by ultrasound. The speed of wave propagation at various junctions, in addition to beam deflection, is the main source of errors in the location of discontinuities. In cases where ultrasonic testing does not detect unacceptable discontinuities (no echoes are obtained), in terms of size and intensity, these discontinuities do not exist in the welds. Nevertheless, in instances where echoes from discontinuities emerge in ultrasonic tests, indicating an unsatisfactory level of joint quality, and the discontinuities are located primarily in its transition zone, radiographic tests should be performed in accordance with the requirements of PN-EN ISO 22825:2017-12.

## 4. Purpose of the Study and Research Methodology

The purpose of this study is to determine the applicability of UT testing to the inspection of welded joints of heterogeneous materials—austenitic materials with ferritic materials. The echo technique and transverse and longitudinal wave heads have been used for this purpose. The 13CrMo4-5 and X2CrNiMo17-12-2 (according to the international designation AISI 316L) steels were selected as the test materials for the study due to their properties and widespread use in the power industry. The scope of the work included the testing of 8, 12, and 16 mm thick welded joints, 300 × 300 mm in size, produced by the 135 (MAG) process using Lincoln 309LSi (Lincoln, Poland) filler material (a welded joint between a ferritic steel component and an austenitic steel component using a ferritic–austenitic filler). The shielding gas used was an M12 mixture containing 97.5% Ar and 2.5% CO_2_.

The various stages of the research task included:Making distance–amplitude curve (DAC) patterns from the test materials;Preparation of specimens of welded joints with artificial discontinuities in the form of through-holes;Performing UT tests on welded joints with artificial discontinuities using heads with 60° and 70° angles for the transverse wave and angle heads for longitudinal waves with similar beam insertion angles;Selection by radiographic testing (RT) of welded joint specimens with natural discontinuities in the form of a lack of sidewall fusion;Performing UT tests on welded joints with natural discontinuities using heads for welded joints with artificial discontinuities.

Low-alloy steel 13CrMo4-5 is a ferritic chromium–molybdenum steel with a relatively low carbon content compared to other related grades used for high-temperature work. It is characterized by good cold and hot formability, very good machinability, and weldability while maintaining high strength properties at both ambient and elevated temperatures. This steel is used to produce parts of fittings for installations—heads, flanges, elbows, tees, profiles, seamless tubes, bars, bushes, forgings, flat bars, and hot-rolled sheets in various thicknesses and cross-sections. The above fittings are used for steam discharge lines and pipelines, as well as for superheater tubes. 13CrMo4-5 steel is mainly supplied in quenched and tempered, softening annealed, and normalized and tempered conditions. The approximate chemical composition of the steel is given in Table 1.

X2CrNiMo17-12-2 (316L) steel is an austenitic steel used for the manufacture of components used in the chemical, paper, automotive, and food industries [39]. It is also used for various types of filters and heat exchangers. It is resistant to intergranular corrosion and acetic and sulfuric acid. It is characterized by high ductility and plasticity and does not change its physical properties at elevated temperatures. The chemical composition of the steel, together with heat treatment conditions and tensile strengths, is given in Table 2.

The most popular welding wire for joining austenitic and ferritic steels is LNM 309LSi, a solid wire with an austenitic–ferritic structure for welding austenitic steels with low-alloy steels. The high Si content improves the wettability. The chemical composition of the wire is given in Table 3, and the mechanical properties are given in Table 4.

The tests were carried out on prepared welds with artificial and natural discontinuities of 8, 12, and 16 mm thickness. Figure 1 shows the elements of the weld groove and the parameters used. The resulting butt joints used a V-type weld groove bevel, which is the most common practice in industrial applications for the MAG welding method. The bevel angle was 25°. The weld thicknesses used reflect those most commonly used for industrial components.

Figure 2 shows the patterns prepared for testing with through-holes drilled perpendicular to the surface of the welded joints to provide a reference line on the screen of the ultrasonic defectoscope. The set of standards used in the tests included welded plates and discs made of the materials to be tested. These were required to set the velocity of the transverse waves and the divergence of the longitudinal waves.

Reference reflectors in accordance with EN ISO 17640 [40], in the form of through-holes 3 mm in diameter and 30 mm in length, drilled perpendicular to the surface of the specimen at various locations on the welded joints were made on reference specimens of 8, 12, and 16 mm thickness. The reference reflectors were made in the weld axis (at the center of the weld, at the top below the surface) in the fusion line and behind the fusion line on both the 13CrMo4-5 and 316L material sides. The performance of the artificial reference reflectors in different zones of the welded joint was intended to verify the feasibility of ultrasonic testing for the considered test areas.

Figure 3 and Figure 4 show a diagram indicating the location of the artificial reference reflectors and photographs of the reflectors, respectively. For measurement purposes, the test plates located to the left and right sides of the welded joint are marked with the symbols A and B.

Tests on the welded joints with artificial discontinuities were followed by tests on welded joints with natural discontinuities. The tested plates with natural discontinuities were characterized by a lack of a sidewall, which is characteristic of MAG-welded joints. Their distribution is shown in Figure 5.

In addition to ultrasonic testing, radiographic and metallographic tests were carried out on weld specimens to confirm that they had been properly prepared. Macroscopic metallographic examinations were carried out to confirm the locations of natural discontinuities detected by UT. The deposits were obtained by cutting the test plates in a vertical plane at the location of the discontinuities using a Bomar saw, shown in Figure 6, at the point where the maximum amplitude value originating from the discontinuity in question was recorded. The surface condition of the metallographic scrap was corrected by milling and grinding. The obtained specimens were then photographed on the bench using an Olympus SZX 10 stereoscopic microscope (Evident/former Olympus/, Tokyo, Japan) equipped with a 0.5 eyepiece and an EP50 camera. The obtained images were graphically processed using the EPview program (Olympus Stream Desktop 2.4) used with the microscope.

As a result of the tests, the prepared macrographic sections showed natural discontinuities in the form of the lack of a sidewall on the walls of the weld groove. Such discontinuities are characteristic of the MAG method used to produce the specimens. Figure 7 shows an example of images obtained on a deposit of a 316L steel welded joint. The arrows indicate the location of the discontinuities. The figures show a cross-section of the whole specimen with discontinuities and zoom in to show the discontinuities more clearly.

Ultrasonic testing was performed with an Olympus EPOCH 650 ultrasonic defectoscope (Olympus/now Evident/, Westborough, MA, USA) with Type A imaging, using AM4R-8x 9-70-4 MHz, AM4R-8x9-60-4 MHz, 70-4 MHz VSY, and 60-4 MHz VSY ultrasonic heads. A mixture of FERDOM’s CH-1 inhibitor (an agent used to protect central heating systems from deposits) and water in a 1:25 ratio was used as the coupling agent.

The choice of beam angle depends on the thickness of the weld to be inspected. For thicknesses from 8 to 15 mm, the relevant standards (for example, EN ISO 17640:2018 [40]) recommend a basic angle of 70° and an additional angle of 60°, while for thicknesses above 15 mm, they recommend a basic angle of 60° and an additional angle of 70°.

## 5. Results of the Study

As a result of preliminary activities before the test, the defectoscope and heads were checked according to PN-EN ISO 22232-3 [41], the speed of the wave in the material was determined, the center of the ultrasonic head was measured, the angle of the introduction of the ultrasonic wave beam into the tested material was determined, and the observation range was selected. Transverse and longitudinal transducers with an angle of 70° were used in the tests. For joints that were 16 mm thick, additional longitudinal and transverse wave heads with an angle of 60° were used.

Holes and discontinuities were detected with a direct incident wave from the mid-stroke when longitudinal waves were used. A summary of the results of the baseline and applied reinforcement for 13CrMo4-5–316L steel using transverse ultrasound waves is shown in Table 5.

The transverse wave inspection included the evaluation of discontinuities at the location of the maximum amplitude of the ultrasonic wave. Firstly, the sensitivity of the test was measured using a reference hole drilled in the weld to allow for attenuation in the welded joint being tested. The baseline gain was then set so that the echo from a 3 mm diameter hole made in the base material was at 0.8H of the screen height (SH). A summary of the baseline and applied reinforcement results for 16 mm thick 13CrMo4-5–316L steel using longitudinal ultrasonic waves is shown in Table 6.

Subsequently, the amplitudes of reflectors, in the form of cylindrical holes located in different zones of the welded joint, were measured, and discontinuities were evaluated using plotted comparison lines. The registration, acceptance, and evaluation levels were determined and the test results were analyzed according to the adopted criteria. According to the accepted evaluation levels, length measurements were carried out for natural discontinuities.

The scanning range during the test was sufficient to cover 100% of the weld volume and the heat-affected zone (HAZ), where a range of 10 mm on each side of the weld is assumed to include this zone. Joints were tested using the same technique—within 1.25 full strokes of the ultrasonic beam for transverse wave heads and 0.5 stroke for longitudinal PN-EN ISO 17640:2019-01 [42]. The tests were performed in accordance with the recommendations of the standard, which required adequate access to both sides of the joint.

The main results of the tests carried out are presented in the following subsections, broken down by type—natural and artificial—and by the transverse and longitudinal test heads used. The following subsections present the values of the changes in decibel gain levels. The results presented were averaged. Each measurement for a given measurement point was taken five times to minimize measurement error. In the case of ultrasonic testing using the echo technique, the assumed accuracy of evaluation that can be assumed in the study is 1 mm. Values in the order of tenths of a millimeter are not possible.

### 5.1. Artificial Discontinuities—Testing with a Transverse Wave Head

Figure 8 shows the results obtained when artificial discontinuities were tested in an 8 mm thick welded joint of dissimilar materials. For this type of joint, reference reflectors were distributed at six test locations, which were measured from the rim and face area of both the ferritic steel side (Figure 8A) and the austenitic steel side (Figure 8B). From the data presented, it can be seen that in the case of ferritic steel, the value of the recorded amplitude for the discontinuity localized at the measurement point of 3-weld root A did not exceed 20% SH. For 316L steel, such a relationship was obtained for two positions: 3-weld root B and 5-weld face B. However, the highest amplitude values were obtained for the 4-weld face B and 6-weld face B reflectors and amounted to 110% SH. For 13CrMo4-5 steel, the highest amplitude value was 101% and was recorded for reflector 5-weld face A. The lowest values were obtained for positions 3-weld root A (15%) and 5-weld face B (17%). In addition, it can be seen that the reference reflector number 3 is poorly detectable for both the A and B sides of the investigation. In the case of the other discontinuities, a certain correlation can be seen, namely, that the measuring points that are better recorded for ferritic steel are less visible for austenitic steel and vice versa.

The amplitudes recorded for a welded joint of dissimilar materials with a thickness of 12 mm are shown in Figure 9 (A—ferritic steel, B—austenitic steel). For 13CrMo4-5 steel, the amplitude did not exceed 20% SH for the discontinuity located at the 2-weld root A. For 316L steel, values below 20% SH were obtained for two positions: 2-weld root B and 4-weld root B. The highest amplitudes were recorded for discontinuities located in the following measurement areas: 4-gradient A (96% SH) and 1-gradient B (101% SH). On the other hand, the lowest amplitude was obtained for reflector number 2, located in the ridge area, for both ferritic (11%) and austenitic (7%) steels. For 13CrMo4-5 steel, reflector number 4 is the best detectable, and number 2 is the weakest. For 316L steel, discontinuities number 3, 5, and 6 have good detectability, while discontinuities number 4 and 2 are slightly worse. Reference reflector number 1 is well detectable from both sides tested.

The amplitudes recorded for a welded joint of dissimilar materials with a thickness of 16 mm are shown in Figure 10 (as before: A—ferritic steel, B—austenitic steel). For 13CrMo4-5 steel, the amplitude value did not exceed 20% SH for the discontinuity located in 2-weld root of A. For 316L steel, values below 20% SH were obtained for two positions: 2-weld root B and 4-weld root B. The highest amplitudes were recorded for discontinuities located in the following measurement areas: 4-gradient A (96% SH) and 1-gradient B (101% SH). On the other hand, the lowest values for both ferritic (11% SH) and austenitic (7% SH) steels were obtained for reflector number 2 located in the ridge region. For 13CrMo4-5 steels, reflector number 4 is the best detectable, while number 2 is the weakest. For 316L steels, discontinuities number 3, 5, and 6 have good detectability, while discontinuities number 4 and 2 are slightly worse. Reference reflector number 1 is well detectable from both sides tested.

Figure 11 shows the values of amplitudes recorded during the test of a 16 mm welded joint measured with a transverse wave head with a beam insertion angle of 60° (A—ferritic steel, B—austenitic steel).

For the two measuring points measured on the side of the 13CrMo4-5 steel, the amplitude value did not exceed 20% SH: 4-weld root A and 6-weld face A. For the 316L steel, only one measuring point achieved a value below 20% SH —5-weld root B. The highest amplitude values of 110% were recorded for discontinuities located at the following measurement positions: 2-weld root A, 5-weld face A, 1-weld root B, 3-weld root B, and 4-weld root B. On the other hand, the lowest values of 14% were obtained for reflectors located in areas 4-weld root A, 6-weld face A, and 5-weld root B. Artificial discontinuities numbered 1, 3, 4, and 6 are better detectable on the austenitic steel side, while 1 and 5 are better detectable on the ferritic steel side. The results are characterized by slightly lower values of the recorded signals compared to those obtained with the 70° beam angle head.

### 5.2. Natural Discontinuities—Testing with a Transverse Wave Head

Figure 12, Figure 13 and Figure 14 show the results obtained for welded joints with natural discontinuities using a transverse wave head with a beam introduction angle of 70° (A—austenitic steel, B—ferritic steel). Compared to the reference reflectors, the amplitudes recorded for natural discontinuities located in welded joints with thicknesses of 8, 12, and 16 mm are lower. From the data obtained for the 8 mm thick plate, it can be seen that in the case of testing discontinuities on the austenitic steel side (Figure 12A), the value of the recorded signal exceeds 20% SH in half of the results. The highest value was recorded for measurement point 2-weld face A (52%), and the lowest for measurement point 3-weld face A (9%). In the case of the tests carried out on the ferritic steel side, only one position recorded an amplitude value above 20% SH (2-weld face B), and this is also the highest value obtained, which is 31% (Figure 12B), while the lowest value was obtained for position 2-weld face B—6%. In addition, no signal was detected for the 1-weld face B and 3-weld face B positions.

Comparing the results of the austenitic and ferritic side measurements, it can be seen that the recorded signals from individual discontinuities are higher for the former. In the case of testing a 12 mm thick plate, only three measurement points tested from the 316L steel side (Figure 13A) obtained an amplitude value above 20% SH (1-weld face A, 2-weld root A, and 3-weld face A). On the other hand, from the measurements made on the 13CrMo4-5 steel side (Figure 13B), it can be seen that the amplitude exceeded the value of 20% SH for only two points (2-weld face B and 3-weld face B). The highest amplitude values were recorded for 2-weld face B, 47% SH, and for 1-weld face A and 2-weld face A the values were 22% SH. On the other hand, the lowest signals were obtained for 2-weld face A (13%) and 3-weld face A (13%) for 316L steel, and 4-weld face B (3%) for 13CrMo4-5 steel. Again, better detectability, i.e., higher amplitudes, were obtained for the measurements conducted on the austenitic steel side. The exceptions are the discontinuities measured in the areas 2-weld face B and 3-weld face B, where higher readings were obtained from the ferritic steel side.

The values of the recorded amplitudes for a 16 mm thick plate are shown in Figure 14. For austenitic steel (Figure 14A), three measurement points were characterized by signal values above 20% SH (1-weld face A, 1-weld root A, 3-weld face A). In the case of ferritic steel, half of the recorded amplitudes exceeded 20% SH. The highest values were recorded for 4-weld face B, 47% SH (13CrMo4-5), and 1-weld face A—26% SH (316L). The lowest amplitudes, on the other hand, were obtained for 3-groove A (8%) and 2-groove B (5%). Natural discontinuities in the measurement areas of 1-weld face B and 3-weld face B were not detected. In addition, as in the case of the study of 8 and 12 mm thick plates, in this case, better detectability was also obtained for discontinuities measured on the austenitic steel side.

Figure 15 shows the results of a test carried out on a 16 mm thick welded joint using a transverse wave head with a beam insertion angle of 60° into the material. From the data presented, it can be seen that, as in previous measurements, the amplitudes of the recorded signals are lower than those obtained for reference reflectors. On the other hand, comparing the values with the data obtained when measuring the same plate, but using a 70°, 4 MHz head, it can be observed that tests performed with a smaller angle of beam introduction allow better detection of natural discontinuities. In the case of ferritic steel (Figure 15B), all the values exceeded 20% SH, whereas in the case of austenitic steel (Figure 15A) such a relationship was obtained for only half of the discontinuities. The highest amplitudes were recorded for 4-weld face A (86%) and 3-weld face B (83%). In contrast, the lowest amplitudes were obtained for discontinuities located in the survey areas 3-weld face A (11%) and 2-weld face B (22%). Unlike previous studies, this time better detectability was obtained for natural discontinuities measured from the ferritic steel side.

### 5.3. Artificial Discontinuities—Tests with a Longitudinal Wave Head

The recorded values of the amplitudes for artificial discontinuities obtained when testing welded joints of various thicknesses of 8, 12, and 16 mm using a longitudinal wave head with a beam introduction angle of 70° are shown in Figure 16, Figure 17 and Figure 18 (A—ferritic steel, B—austenitic steel).

In the case of an 8 mm thick welded joint and measurements carried out on the side of 316L steel, it can be seen that all the values obtained exceeded 20% SH (Figure 16B), whereas in the case of 13CrMo4-5 steel, the value of the amplitude for measurement point 6-weld root A did not exceed this (Figure 16A). The highest values were recorded for the reflectors located at 2-weld face A (79%) and 6-weld face B (94%), and the lowest for 6-weld face A (13%) and 3-weld face B (23%). When comparing the individual values of the amplitudes recorded for the reflectors during their examination on the ferritic and austenitic steel sides, a slightly better detectability was observed on the 13CrMo4-5 steel side.

From the data obtained for a 12 mm thick plate, it can be seen that the values of the amplitudes obtained from the examination of the discontinuities on both sides of the joint exceed 20% SH (Figure 17). For ferritic steel, the highest value was recorded for the reflector in the point 5-weld face A area (76%), and the lowest value of 27% was obtained at two measurement points: 3-weld face A and 6-weld face A. For austenitic steel, the highest value was 82% (2-weld face B) and the lowest was 20% (3-weld face B). After comparing the amplitudes recorded for individual reflectors on both sides of the joint, no significant differences in their detectability were observed. All the amplitudes obtained during the test of the 16 mm thick plate also exceeded 20% SH (Figure 18). The highest values of 73% and 88% were recorded from the following tested areas 1-weld face A, 2-weld face A, and 4-weld face B. On the other hand, the lowest values were recorded from the points 4-weld face A (20%) and 3-weld face B (40%). Reflector number three was not detected on either side of the face. As in the case of the test of the 8 mm thick plate, there were no significant differences when comparing the values of the amplitudes obtained from the measurements on the ferritic and austenitic steel sides.

On the other hand, when comparing the data obtained from the transverse wave head test with those obtained from the measurements made with the longitudinal wave head, no clear differences could be observed in the amplitudes recorded for each reference reflector. In the case of the 8 and 12 mm plates, a slightly better detection of discontinuities was obtained when testing with the longitudinal wave head, while for the 16 mm plate, with the transverse wave head.

The results recorded for a 16 mm plate with a 60° longitudinal wave head are shown in Figure 19 (consistently: A—ferritic steel, B—austenitic steel). All the amplitudes recorded on the austenitic steel side are above 20% SH, while for ferritic steel only one value is below this level. The highest amplitude value of 110% SH was obtained for the following measurement areas: 1-weld face A, 4-weld face A, 2-weld face B, 3-weld face B, and 3-weld face B. In contrast, the lowest values were obtained for 6-weld face A (18%) and 5-weld face B (20%). Better detectability for most artificial discontinuities was recorded for measurements on the austenitic steel side. However, if the results are compared with those obtained using a 70° head, it is clear that the detection of discontinuities is better at a smaller angle. Measurements made at the same angle but with a transverse wave head gave higher values of recorded amplitude signals when tested with a longitudinal wave head.

### 5.4. Natural Discontinuities—Tests with a Longitudinal Wave Head

The results obtained from measurements of welded joints containing various natural discontinuities are shown in Figure 20, Figure 21 and Figure 22 (A—austenitic steel, B—ferritic steel).

Based on the values obtained during the testing of an 8 mm thick welded joint (Figure 20), it was observed that the amplitude values for the discontinuities of 1-weld root A, 3-weld root A, and 4-weld root A tested on the 316L steel side did not exceed 20% SH. However, the highest value was recorded for the position of 1-weld face A (69%), and the lowest value of 16% was recorded for 1-weld root A and 3-weld root A. On the other hand, for the ferritic steel, an amplitude value of less than 20% SH was obtained for 4-weld face B, and this is also the measurement point with the lowest amplitude (18%). The highest signal value of 40% was obtained for discontinuities number 3 and 4 measured from the ridge area. The detectability of individual discontinuities from both sides of the tested joint was similar; there were no significant differences in the values of the recorded amplitudes.

In the case of the data obtained from measurements on a 12 mm thick plate, it was observed that only the signal amplitude of discontinuity number 4 measured from the rim area did not exceed 20% SH from both the 316L and 13CrMo4-5 steel sides (Figure 21). For measurements taken on the austenitic steel side, the highest amplitude value of 38% was obtained for 1-weld face A and 2-weld face A, while on the ferritic steel side it was obtained for 3-weld root B (64%). In contrast, the lowest values were recorded for discontinuities number four from the ridge area, where they were 17% and 15% for 316L and 13CrMo4-5 steels, respectively. As with the 8 mm thick plate, there is no significant difference in the detectability of individual discontinuities depending on the side of the test. The amplitude values recorded during the measurement of a 16 mm thick welded joint are shown in Figure 22.

All the amplitude values obtained during the measurement on the 316L steel side were above 20% SH, whereas on the 13CrMo4-5 steel side, for the discontinuity located in the 3-weld root B area, the value was below this at 13%. At the same time, this represents the least detectable discontinuity for measurements from this side of the joint. On the other hand, for the tests carried out on the austenitic steel side, the lowest value was recorded for 2-weld root A. The highest values were observed for 4-weld root A, 45% (316L), and 4-weld root B, 60% (13CrMo4-5). Comparing the recorded amplitudes obtained during the longitudinal wave head test with those obtained during the transverse wave head measurements, it was observed that in the case of the former, the detection of individual natural discontinuities was better for all the welded joint thicknesses tested.

Figure 23 shows the results obtained when testing a 16 mm thick plate with a 60° beam angle longitudinal wave head (A—austenitic steel, B—ferritic steel). Compared to the tests carried out with the 70° head, a deterioration in the detection of individual discontinuities can be seen, namely, for three of the tested areas on the 316L steel side and for half on the 13CrMo4-5 side, the value of the amplitudes did not exceed 20% SH. The highest signal value of 104% was recorded for 1-weld root A and 4-weld root B, while the lowest values were recorded for 4-weld root A (13%), 2-weld face B (16%), and 2-weld root B (16%). There were no significant differences in the results obtained when testing at the same angle but with a transverse wave head in the detection of individual natural discontinuities.

### 5.5. Values of Changes in Decibel Gain Level—Transverse Wave Head

Table 7 and Table 8 show the values of the change in decibel gain level obtained when recording the signal from the discontinuity in relation to the DAC reference line. The results obtained when testing the reference reflectors with the transverse wave head are shown in Table 7. From these data, it can be seen that for the 8 mm thick test welded joint the change in the decibel gain level ranges from −10.4 to 6 dB for ferritic steel, and from −6.4 to 4.5 dB for austenitic steel. For a 12 mm thick plate, the ranges are −9.5 to 1.6 dB and −14.5 to 6.8 dB, respectively. The ranges of change in decibel gain levels obtained when testing a 16 mm thick welded joint are −0.6 to 4.3 dB (13CrMo4-5) and −9.4 to 7.6 dB (316L). For the data obtained when testing a 16 mm plate but at a smaller beam angle, the ranges vary from −15.3 to 6.2 dB and −5.7 to 9.3 dB, respectively.

Tests for artificial reference discontinuities distributed over different areas of the welded joint were carried out on the face and rim sides using transverse wave heads, and the measurement points were then classified according to the acceptance criteria adopted (Table 7).

For the 8 mm thick joint, the results exceed the rating level of −14 dB, and allow the evaluation of all recorded artificial discontinuities for both the 13CrMo4-5 and 316L steel side tests. The lowest result was recorded for test point 4-weld root A (−10 dB). However, this exceeded the registration level of −14 dB and was equal to the acceptance level of −10 dB, so it was unacceptable, as were the other measurements made on the measurement points as artificial reference reflectors. In each case, the −10 dB acceptance level was exceeded.

For the 12 mm thick welded joint, a result below the evaluation level of −14 dB was obtained for the reading from measurement point 4-weld root B (−14.5 dB). According to the specified criteria, this result from the 316L steel side should not be taken into account. The remaining results were unacceptable and exceeded the acceptance level of −10 dB.

For the tests carried out on the welded joint with a thickness of 16 mm using 70° and 60° beam introduction angles, all the indications from the reference discontinuities exceeded the −14 dB evaluation level. They were simultaneously indications that exceeded the level of registration and acceptance, and were therefore unacceptable in view of the assumptions made.

A similar analysis of the data was carried out for the decibel gain level values obtained when natural discontinuities in welded joints were detected using a transverse wave head (Table 8). In this case, the ranges of decibel level changes for ferritic steel were −15.4 to 0 dB for 8 mm, −20.6 to −12.9 dB for 12 mm, −24.8 to 6.8 dB for 16 mm, and −6.8 to 5.7 dB for 16 mm, at a 60° angle. For austenitic steel, the ranges were −14.8 to −1.8 dB, −12 to −4.7 dB, −12.5 to −2.6 dB, and −11.7 to 2.1 dB, respectively. It should also be noted that these values were negative for austenitic steel when tested with a head with a beam introduction angle of 70°. For ferritic steel, such a relationship was only obtained when testing a 12 mm thick joint.

The tests for natural discontinuities carried out with a transverse wave head with the same reinforcement as for artificial reflectors gave the following results according to the adopted evaluation criteria (Table 8).

For a thickness of 8 mm, the measurement points 1-weld root B (−15 dB), 2-weld root B (−15.4 dB), and 3-weld root A (−14.8 dB) were below the −14 dB evaluation level and were not considered. Indications from the measurement points 1-weld face A (−7.8 dB), 2-weld face A (−1.8 dB), 2-weld face A (−3 dB), and 3-weld face A (−4.3 dB) were unacceptable indications, exceeding the −14 dB registration level and the −10 dB acceptance level. Measurement points below the −14 dB registration level but above the −10 dB acceptance level were recorded and were acceptable according to the imposed criteria.

For a welded joint with a thickness of 12 mm, eight measurement points were unclassifiable, exceeding the −14 dB evaluation level. It is noteworthy that all of these locations were on the test side of the 13CrMo4-5 material. The discontinuities detected on the 316L steel side, with the exception of the reading from measurement point 3-weld face A (−12 dB), were unacceptable as they exceeded the −10 dB acceptance level.

Tests carried out on a 16 mm thick welded joint and using an angle of 70°, with the exception of the indication of 2-weld face B (−24.8 dB), were indications considered for evaluation and subject to registration. Indications exceeding the level of registration but not exceeding the level of acceptance were recorded for measurement points 2-weld face A (−10.2 dB), 2-weld root A (−12 dB), 2-weld root B (−12, 2dB), and 3-weld root A (−12.5 dB). Other indications from discontinuities were unacceptable.

The use of a 60° angle resulted in indications exceeding the −14 dB evaluation level. The indications subject to registration and unacceptable discontinuities were those from the measurement points 2-weld root B (−11.7 dB) and 3-weld face B (−11.2 dB). The remaining discontinuities were unacceptable according to the imposed criteria.

The range of the ultrasonic wave path obtained when measuring the 8 mm thick welded joint with artificial reference reflectors using a transverse wave head with 70° beam angle introduction and made of 13CrMo4-5–316L steel was from 14.9 to 62.2 mm for ferritic steel, and from 25.6 to 64.1 mm for austenitic steel (Table 7). For the 12 mm thickness, the range was from 18.6 to 65.4 mm and from 18.2 to 63.4 mm, respectively. For the 16 mm thick welded joint tested with a 70° head, the path ranges were 23.8 to 91.8 mm and 22.1 to 74.1 mm, respectively, and for a 60° head, 14.3 to 78.1 mm and 13.9 to 74.6 mm, respectively.

The path ranges for welded joints with natural discontinuities, measured with the transverse wave head, are shown in Table 8. For the 8 mm thick plate of 13CrMo4-5–316L steel, they ranged from 10.0 to 57.7 mm for ferritic steel, and from 13.3 to 57.8 mm for austenitic steel; for the thickness of 12 mm, the ranges were from 18.7 to 80.5 mm and from 31.6 to 60.6 mm, respectively. In the case of the 16 mm thick welded joint, tested with a 70° head, the path ranges were from 24.2 to 71.7 mm, and from 22.1 to 74.1 mm, respectively. For the 60° angle head, the values were from 16.8 to 55.5 mm and from 20.9 to 49.6 mm, respectively.

### 5.6. Values of Changes in Decibel Gain Level—Longitudinal Wave Head

The values of decibel gain level changes obtained by recording the amplitude signal against the DAC curve using the longitudinal wave head are shown in Table 9 (for artificial discontinuities) and Table 10 (for natural discontinuities). From the data in Table 9, it can be seen that when testing the 8 mm thick welded joint, the range of decibel gain varied from −15.3 to 0.4 dB for 13CrMo4-5 steel and from −9.2 to 2.5 dB for 316L steel. For the 12 mm joint, the ranges were −8 to 2.2 dB and −8.8 to 2.1 dB, respectively, while for the 16 mm joint, the ranges were −7.5 to 2.6 dB and −1.4 to 3.4 dB, respectively. When the 16 mm plate was tested using a head with a smaller beam angle (60°), the ranges of changes in decibel gain were −10.3 to 9.8 dB (ferritic steel) and −17.3 to 5.5 dB (austenitic steel).

The results obtained during the tests for welded joints with artificial discontinuities with the longitudinal wave head were also verified according to the adopted evaluation criteria (Table 9). For the thickness of 8 mm, the value of the decibel gain recorded for measurement point 6-weld root A (−15.3 dB) was below the evaluation level of −14 dB. The remaining values obtained for individual measurement points were evaluated, as was the case for the 12 mm thick welded joint, where all values met the criteria. When the results were considered in relation to the acceptance level of −10 dB, all indications were unacceptable. For the joint with a thickness of 16 mm and an angle of 70°, all measurement values exceeded the evaluation level of −14 dB while simultaneously exceeding the registration and acceptance levels, and were therefore unacceptable. On the other hand, in the case of measurements carried out for an angle of 60°, measurement point 5-weld root B (−17.3 dB) was not evaluable, and 6-weld root A (−10.3 dB) was a registrable indication and met the acceptance level. The remaining artificial reference discontinuities did not meet the acceptance levels.

For the natural discontinuity test (Table 10), the ranges of change in the decibel gain level of the recorded signal relative to the reference line for ferritic steel were −12 to −3.6 dB (8 mm), −11.5 to 0.9 dB (12 mm), −15.1 to 0.3 dB (16 mm), and −11.8 to 7.4 dB (16 mm and 60° angle). For austenitic steel, the ranges were −11.8 to 7.4 dB, −9.5 to −1.6 dB, −8.3 to −1.8 dB, and −15.7 to 5.5 dB, respectively. As with the study of natural discontinuities using the transverse wave head, negative ranges were also obtained, but only for the austenitic steel specimens with 12 and 16 mm thickness.

As for artificial discontinuities, tests for natural discontinuities were carried out on the side of the 13CrMo4-5 and 316L materials, which were then verified according to the adopted evaluation criteria (Table 10).

All indications of discontinuities for the 8 and 12 mm thicknesses of the heterogeneous joints were subject to evaluation because they exceeded the evaluation level of −14 dB. Discontinuities subject to registration but not exceeding the acceptance level, were indications from measurement points 1-weld face A (−11.8 dB), 4-weld face B (−12 dB), and 4-weld face A (−10.7 dB) for the 8 mm thickness; and 4-weld face B (−11.5 dB) for the 12 mm thickness. The remaining discontinuities did not meet the requirements for the imposed acceptance levels. For tests performed on the 16 mm thick welded joint, only the indication from the 3-weld root B measurement point (−15.1 dB) was below the evaluation level. The remaining discontinuities exceeded the registration and acceptance levels and were classified as unacceptable. For the tests performed on the 16 mm thick specimen using a 60° angle head, only the discontinuity at measurement point 4-weld root A (−15.7 dB) was not evaluated. Measurement points 1-weld face A (−10.7 dB), 2-weld face B (−11.8 dB), 2-weld face A (−11.4 dB), 2-weld face B (−11.8 dB), and 3-weld face B (−10.9 dB) were above the registration level but below the acceptance level. The remaining discontinuities were below the acceptance level.

Table 9 and Table 10 summarize the ultrasonic wave path values obtained from the measurements carried out with the longitudinal wave head on welded joints containing artificial and natural discontinuities.

The path range for the 8 mm thick plate made of 13CrMo4-5–316L steel with reference reflectors was from 19.6 to 29.8 mm for ferritic steel, and from 15.5 to 30.0 mm for austenitic steel. For the 12 mm plate, the ranges were 16.3 to 33.4 mm and 16.7 to 33.8 mm, respectively. For the 16 mm thick welded joint tested with a 70° head, the path ranges were 22.3 to 45.3 mm and 23.0 to 41.6 mm, and for a 60° head, 12.7 to 34.1 mm and 14.9 to 35.2 mm, respectively.

The path range for measuring a welded joint with natural discontinuities using a 70° longitudinal wave head and an 8 mm thickness was from 20.8 to 32.9 mm for ferritic steel, and from 28.5 to 32.2 mm for austenitic steel. For the 12 mm thick plate, the ranges were 26.9 to 32.6 mm and 28.5 to 42.5 mm. For the 16 mm thick joint, the path range was from 23.3 to 40.1 mm and from 20.6 to 38.9 mm, respectively. The results of measuring the same specimen but with a 60° head gave the ranges of 16.0 to 33.0 mm and from 17.0 to 33.8 mm.

## 6. Evaluation of Ultrasonic Test Results for a Heterogeneous Joint in 13CrMo4-5–316L Steel

In order to determine how the angle of beam introduction into the material or the type of wave used affects the results, a comparative analysis was carried out and the values obtained are shown graphically in Figure 24, Figure 25, Figure 26 and Figure 27. The data are presented for a variety of joints to further analyze the differences in amplitude values recorded from the ferritic and austenitic steel sides. All the amplitude values recorded from the reference holes were divided into three groups, values less than 40% SH, values from 40 to 80% SH inclusive, and values greater than 80% SH, and then, counted and related to their total number. In addition, within each of these ranges it was analyzed how much data was recorded from the face or root area and whether the test was performed from the A side (ferritic steel) or B side (austenitic steel).

Figure 24 shows that the values of the amplitudes increase as the thickness of the tested joint increases. For a joint thickness of 8 mm, the interval containing amplitudes below 40 is more than 50% of all recorded indications, while for a connector thickness of 16 mm, it is only 25%. In the case of this joint, the use of a 60° angle head reduced the number of recorded signals. On the other hand, the use of a longitudinal wave head for testing significantly improved the quality of the signals obtained.

Figure 25 shows that the range of values below 40% SH has been significantly reduced in favor of the range of values between 40 and 80% SH. However, the range of values above 80% SH does not exceed 10% in any of the joints tested. The use of a head with a beam angle of 60° to measure the 16 mm thick specimen improved the quality of the recorded signals, and for more than 50% SH of the amplitudes, the value exceeded 80% SH. From the data presented in the text boxes in Figure 24 and Figure 25, it can be seen that most of the indications characterized by amplitude values above 80% SH were recorded from the face area. This effect is most evident for the longitudinal wave when testing the joints of 8 and 12 mm thickness. However, when analyzing the side from which the highest amplitude values were recorded (Figure 26 and Figure 27), it can be seen that it is the B side, i.e., the austenitic steel side. Furthermore, in the case of the longitudinal wave, except for the beam angle of 60°, all values above 80% SH were recorded from the B side.

Table 11 shows the locations of the measurement points corresponding to different locations of the reference reflectors placed in one of the zones of the welded joint. For each point and all the thicknesses and angles tested, as well as the type of wave, the degree of detection of the artificial discontinuity was determined graphically. On the basis of the results obtained, some correlations were observed. In particular, for the reflector placed in the center of the weld (no. 1), only in the case of the specimen of 8 mm thickness for side B was the area was poorly detectable, and the amplitude was less than 40% SH.

This point was best detected for thicker welded joints and measurements with a longitudinal wave head and an angle of 60°. For the reflector located at the top of the weld under the face (no. 3), a deterioration in detectability was observed. This is particularly noticeable at a joint thickness of 8 mm and other joints when examined from the face area. The reflectors numbered 2 and 4 located on the fusion lines are best detected at a joint thickness of 16 mm using a transverse wave head and an angle of 70° or a longitudinal wave head but an angle of beam insertion into the material of 60°. For reflectors located behind the fusion line on both sides of the tested joint (no. 5 on the ferritic steel side and no. 6 on the austenitic steel side), no improvement in detectability was observed with increasing plate thickness. However, the change in wave type affected the increase in detectability of reflector no. 5, while reflector no. 6 was characterized by significantly higher readings obtained during the test from the face area on the austenitic steel side.

From the summary of the number of points in a given numerical range shown on the right side of Table 11, the most and least detectable artificial discontinuities were determined. Point 1 is characterized by the lowest number of indications of less than 40% SH, with which it is best detected in all the joint thicknesses tested and at both heads and angles (values are marked in green). On the other hand, discontinuities 3, 5, and 6 are the least detectable, with the highest number of indications detected in the range below 40% SH (values marked in red). In addition, it can be seen that these points have the fewest indications of more than 80% SH. A comparable summary has been made in the bottom rows of Table 11, this time counting the points contained in each column. From these data, it can be seen that the least detectable discontinuities are those located in the 8 mm thick joint when using the transverse wave head. On the other hand, when using the longitudinal wave head, and analyzing the number of indications in each interval, very good detection of discontinuities is observed in all the joint thicknesses tested.

## 7. Discussion

Tests conducted on artificial discontinuities in the thickness ranges of 8, 12, and 16 mm using a 70° angle of the transverse wave were best detected at the 16 mm thickness both measured from the austenitic and ferritic steel sides. Differences in the detectability of individual artificial discontinuities depending on the side tested were not as apparent as for other thicknesses. The lowest readings were obtained at a thickness of 12 mm and point 2 face B and 4 root B, where the values of readings did not exceed 10%. Hole 1 was detectable at levels above 50% in all cases from both ferritic and austenitic steels. Hole 3 was the least detectable at the 8 mm thickness on both the ferritic and austenitic steel sides.

The results were similar for each of the materials tested. Reflector 6 face B for each of the thicknesses considered in the 8–16 mm range had higher readings than 6 face A. Reflector 5 face A showed a similar magnitude of recorded amplitude at thicknesses of 8 and 12 mm while for the 16 mm thickness the measured reading indicated a value of 110%. On the B side, the highest values on both the face and border sides were registered at the 12 mm thickness and similar results were registered at the 8 and 16 mm thicknesses and did not exceed the value of 35%.

The use of angle heads for transverse and longitudinal waves gave positive results during the performed tests for artificial discontinuities. In the case of artificial discontinuities, the highest amplitude values were recorded for the 16 mm thick heterogeneous joint using a transverse wave head and an angle of 70° and 60°. For the tests performed on the artificial discontinuities located in the 8 mm thick welded heterogeneous joint with longitudinal waves, higher amplitude values were obtained on the 316L steel side than for the discontinuities measured on the 13CrMo4-5 steel side. For the other thicknesses, of 12 and 16 mm, amplitude values were observed from measurements on both sides of the joint. The highest values of the recorded amplitude were obtained during the measurement for the discontinuities of artificial longitudinal waves and welded joints of different thicknesses of 16 mm and angle of 60°.

Angle heads for transverse waves gave lower amplitude ratios for natural discontinuities than for artificial discontinuities. Not all measurement points could be located. This was the case for point 1 face B with thicknesses of 8 and 16 mm, and 3 root B (thickness 8 mm) and 3 face B with a thickness of 16 mm. For the 16 mm thickness, the readings on each side of the test gave results of more than 10%. No such relationship was observed for the 8 and 12 mm thicknesses. The highest reading rates were obtained for the 16 mm thickness and lower values were registered for the 8 and 12 mm thicknesses.

Tests carried out on representative welded joints in the 8–16 mm range for a longitudinal wave and an angle of 70° with implemented natural discontinuities showed registered amplitude magnitudes higher than for samples with artificial discontinuities. Discontinuities at measurement points 1 face A and 8 mm thick, 3 root B and 12 mm thick, and 4 root B for 16 mm thick gave results of 60% or higher. All other measurement points gave lower results. In the considered thickness interval of 8–16 mm, all measurement points gave a measurement result above 10%. The most similar distribution of amplitudes in the 20–40% range was obtained for measurement points for thicknesses of 12 and 16 mm and tests from test area A. Similar magnitudes were recorded here at a thickness of 8 mm, however, when testing from area B of the welded joint. The use of a 60° angle and a longitudinal wave head gave higher measurement readings for points 3 root B and 4 root B than the use of a 70° longitudinal wave head in this case. The use of the 60° head gave higher magnitudes of measurement readings than the 70° head for points 1 root A and 1 root B, 2 face A, and 3 face A. Point 4 from the root A area gave a higher result for the 70° head than for the 60° angle.

Longitudinal waves propagate at a higher speed than transverse waves. They are less sensitive to deflections at grain boundaries and to obtaining readings derived from apparent occurring indications; i.e., indications that do not originate from natural discontinuities, which are often the result of technological errors in the execution of welded structures.

Normal heads for longitudinal waves are insufficient for ultrasonic testing in heterogeneous joints, so it is necessary to use heads that introduce longitudinal waves at a specific angle. When using dual heads for longitudinal waves, it is important to separate the transmitting and receiving signal paths. This phenomenon does not occur when testing with standard transverse wave heads. The most common use of dual heads is characterized by a better signal-to-noise ratio and a limited sensitivity zone. The narrowing of the sensitivity zone is the result of a longer dead zone with single-angled longitudinal wave heads than with the same transverse wave heads. This is also due to the higher structural noise when testing welds made of dissimilar materials. Transverse waves propagate at a smaller angle and at a slower speed than longitudinal waves. Echoes obtained from longitudinal waves can be identified by their position in a suitably close range of the time base. Interpretation of the indications is limited to half a pitch. In contrast, indications obtained from transverse waves will always be characterized by longer delays. There is, however, a possibility here to perform half-pitch and multiples of it, which gives the possibility of reading from further ranges of the path of the transducer head.

## 8. Conclusions

The essence of the article is to determine the reliability of test results using technique no. 1 according to DAC curves and to analyze the size of the error based on the evaluation of the height of the echo amplitude. The tests carried out on heterogeneous welded joints in this study were exploratory in nature. They have made it possible to formulate conclusions that will inform further consideration of the use of ultrasonic testing to measure heterogeneous joints. It is also a starting point for planning the next stages of testing on, among other things, specimens in the form of plates with other combinations of base and filler materials and the use of other welding processes that affect the change in the structure of welded joints.

The following observations can be made from the results of the tests:For artificial discontinuities, the highest amplitude values were recorded for a 16 mm thick heterogeneous joint using a transverse wave head and angles of 70° and 60° (Figure 8, Figure 9, Figure 10 and Figure 11).For tests performed on natural discontinuities in heterogeneous joints using transverse waves, discontinuities on the side of 316L steel were detected more favorably. On the ferritic 13CrMo4-5 steel side, not all discontinuities were detected (Figure 12 and Figure 15).Testing of a 16 mm thick natural joint using a transverse wave head with a beam insertion angle of 60° had the most favorable detection of natural discontinuities concerning the other measurements made for heterogeneous joints (Figure 12, Figure 13, Figure 14 and Figure 15).For the tests carried out on the artificial discontinuities located in the 8 mm thick variegated welded joint with longitudinal waves, higher values of amplitudes were obtained on the 316L steel side than for the discontinuities measured on the 13CrMo4-5 steel side—Figure 16. For the other thicknesses of 12 and 16 mm, similar values of amplitudes obtained from measurements on both sides of the joint were observed—Figure 17 and Figure 18.The highest values of the recorded amplitude were obtained during the measurement of artificial discontinuities using longitudinal waves and welded joints of various thicknesses of 16 mm and an angle of 60°—Figure 19.No significant differences were observed for tests conducted using the longitudinal wave head on the ferritic and austenitic steel sides at natural discontinuities for both the thickness and the side from which the test was conducted (Figure 20, Figure 21, Figure 22 and Figure 23).Similar ranges of decibel gain level change values to the comparison line carried out on artificial discontinuities in heterogeneous joints using both transverse and longitudinal waves were recorded (Table 7 and Table 9).Higher values of decibel gain level changes to the determined comparison lines were observed when tested on natural discontinuities on the 13CrMo4-5 side of the material at thicknesses of 8 and 12 mm than on the 316L side (Table 8), while smaller differences were observed for tests conducted using longitudinal waves (Table 10).

Based on the study, it is concluded that, despite the difficulties, ultrasonic testing allows volumetric inspection of welded joints of diverse materials of the type of ferritic steel in the grade 13CrMo4-5 with austenitic steel 316L. In addition:The sensitivity of detection in the testing of heterogeneous joints depends on the type of material and its phase structure, the ultrasonic wave used, and the angle of beam introduction.The highest detection sensitivity was obtained by conducting tests from the ferritic steel side, which is due to the lower attenuation of the ultrasonic wave propagating in ferritic steel compared to austenitic steel.The most favorable conditions for detecting discontinuities were obtained when testing using a longitudinal ultrasonic wave.For testing materials up to 15 mm thick, more favorable detection results were obtained with an ultrasonic beam insertion angle of 70° and above 15 mm with an ultrasonic beam insertion angle of 60°.The ultrasonically tested heterogeneous joints required more sweeps for maximum detection of different areas.

Main test results, stated benefits, and limitations:The DAC technique can be used to evaluate heterogeneous welded joints.The value of the recorded amplitudes depends on the location of the reference reflectors.From the conducted tests, the effect of the test side on the registration of the signal from artificial and natural discontinuities was observed.In ultrasonic testing of welded joints of austenitic steels, due to the structural changes in the tested materials, tests should be carried out from each side.When testing austenitic steel, it is recommended to use longitudinal wave heads. They show lower values of the attenuation coefficient, and therefore, smaller corrections to the amplification used when recording the signal from artificial and natural discontinuities.

Directions for further research:Application of phased array technique—this technique uses multi-transducer mosaic heads for testing and uses a wide angular range of 40–70°. The dual matrix array with transmit–receive longitudinal (DMA-TRL) head, which allows the introduction of creeping L-wave, longitudinal L-wave, and the use of the tandem LLT technique, is also becoming increasingly popular.Testing of welded joints with larger thicknesses, in the range of 20–35 mm—these tests use transverse and longitudinal wave heads with frequencies lower than those used within the framework of this work and with other transducer sizes.Testing of welded joints with thicknesses in the range of 3–8 mm—in the energy industry, tests are also performed on joints with smaller thicknesses in the range of 3 to 8 mm using the Cobra technique. Power blocks on heat exchangers, among others, also include connections in this range.Testing of components with a different configuration—welded joints in different combinations of base and filler materials. The use of other welding processes such as TIG, covered electrode, or powder wire welding can also be considered.

## Figures and Tables

**Figure 1 materials-17-04216-f001:**
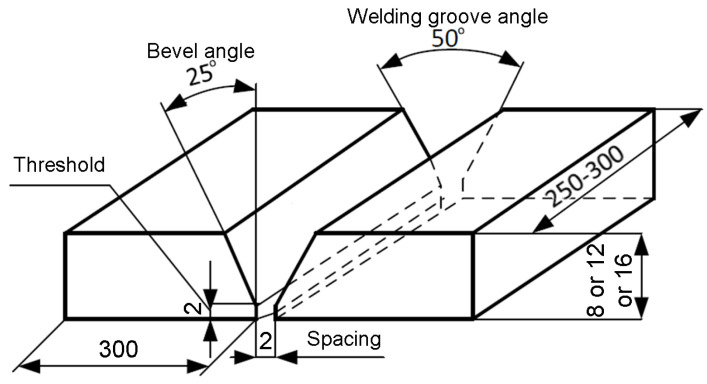
Welding groove components.

**Figure 2 materials-17-04216-f002:**
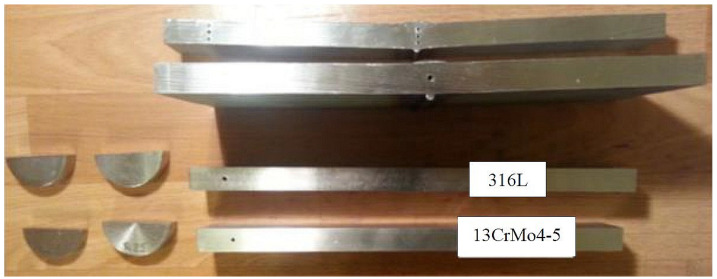
Patterns made from the materials under test.

**Figure 3 materials-17-04216-f003:**
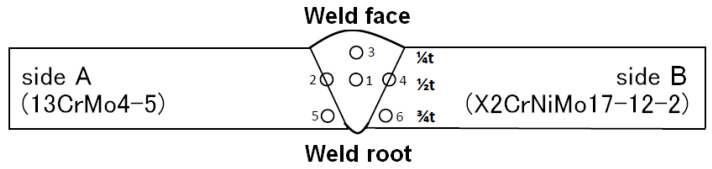
Diagram of the arrangement of artificial gauge reflectors in the form of through-holes for welded joints of dissimilar materials (side A—13CrMo4-5 steel, side B—316L steel); position points: 1—in the center of the weld, 2—on the fusion line on the side of the 13CrMo4-5 material, 3—on the top of the weld under the face, 4—on the fusion line on the side of the 316L material, 5—behind the fusion line on the side of the 13CrMo4-5 material, 6—behind the fusion line on the side of the 316L material.

**Figure 4 materials-17-04216-f004:**
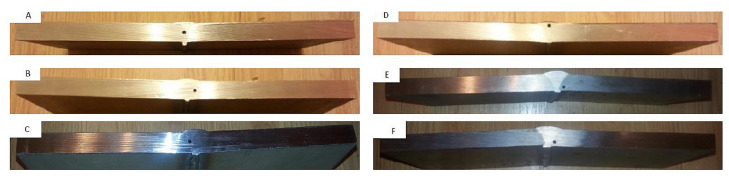
View of the distributed artificial reference reflectors in different zones of the welded joint in the example of a welded joint made of 16 mm thick dissimilar materials: (**A**) in the axis at the center of the weld, (**B**) in the fusion line from the 316L material, (**C**) in the fusion line from the 13CrMo4-5 material, (**D**) in the weld axis at the top of the welded joint, (**E**) behind the fusion line from the 13CrMo4-5 material, (**F**) behind the fusion line from the 316L material.

**Figure 5 materials-17-04216-f005:**
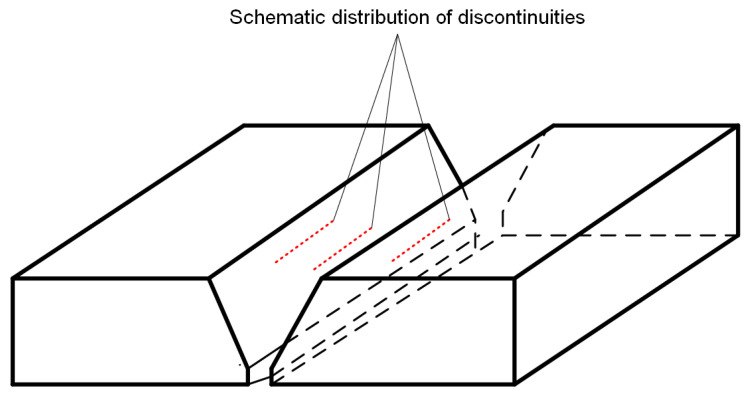
Schematic distribution of discontinuities in fabricated welded joints.

**Figure 6 materials-17-04216-f006:**
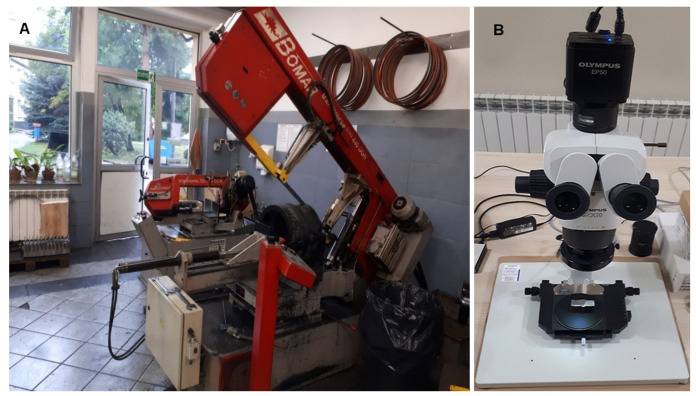
Instruments used to prepare samples and perform metallographic tests: (**A**) a Bomar saw model transverse 610.440 DGH used to cut the plates and (**B**) a station for taking photographic images of the scraps taken (Olympus SZX 10 stereoscopic microscope with EP50 camera).

**Figure 7 materials-17-04216-f007:**
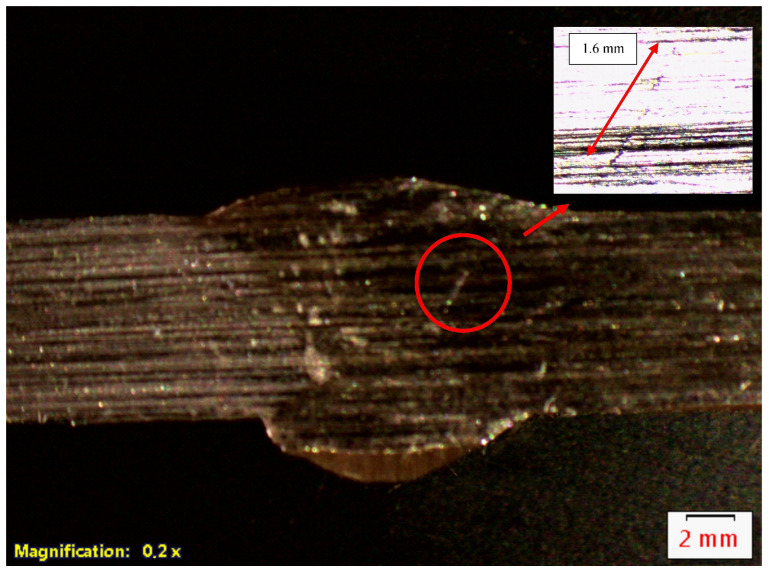
Sample result of macroscopic metallographic examination—welded joint with natural discontinuities; 316L steel with a thickness of 8 mm.

**Figure 8 materials-17-04216-f008:**
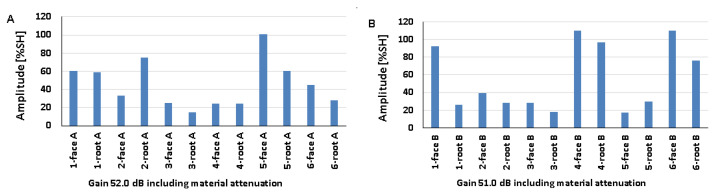
The change in signal amplitude depending on the position of the reference reflectors for (**A**) 13CrMo4-5 and (**B**) 316L materials with a thickness of 8 mm. The test was carried out with a 70°, 4 MHz transverse wave head.

**Figure 9 materials-17-04216-f009:**
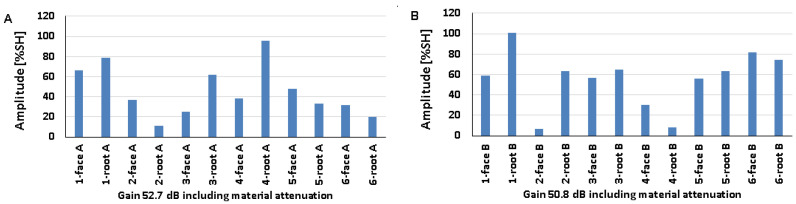
The change in signal amplitude depends on the position of the reference reflectors for (**A**) 13CrMo4-5 and (**B**) 316L materials with a thickness of 12 mm. The test was carried out with a 70°, 4 MHz transverse wave head.

**Figure 10 materials-17-04216-f010:**
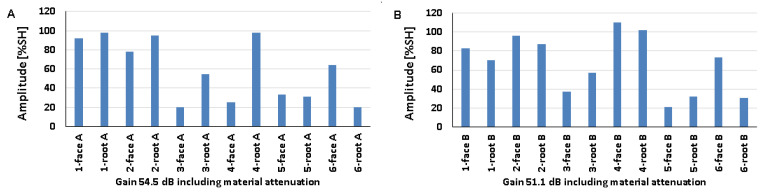
The change in signal amplitude depending on the position of the reference reflectors for (**A**) 13CrMo4-5 and (**B**) 316L materials with a thickness of 16 mm. The test was carried out with a 70°, 4 MHz transverse wave head.

**Figure 11 materials-17-04216-f011:**
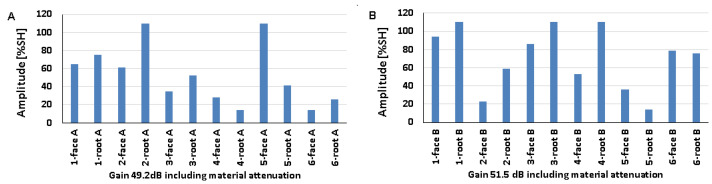
The change in signal amplitude depending on the position of the reference reflectors for (**A**) 13CrMo4-5 and (**B**) 316L materials with a thickness of 16 mm. The test was carried out with a 60°, 4 MHz transverse wave head.

**Figure 12 materials-17-04216-f012:**
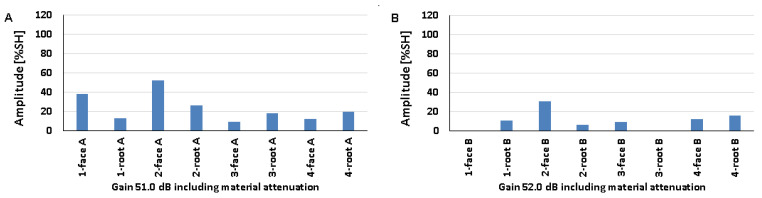
The change in signal amplitude depending on the position of the natural discontinuity for (**A**) 316L and (**B**) 13CrMo4-5 materials with a thickness of 8 mm. The test was carried out with a 70°, 4 MHz transverse wave head.

**Figure 13 materials-17-04216-f013:**
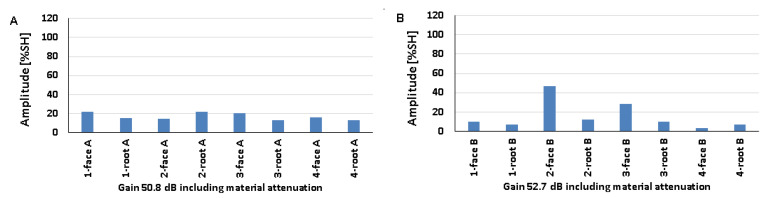
The change in signal amplitude depending on the position of the natural discontinuity for (**A**) 316L and (**B**) 13CrMo4-5 materials with a thickness of 12 mm. The test was carried out with a 70°, 4 MHz transverse wave head.

**Figure 14 materials-17-04216-f014:**
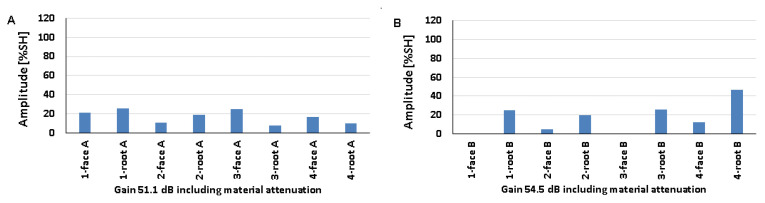
The change in signal amplitude depending on the position of the natural discontinuity for (**A**) 316L and (**B**) 13CrMo4-5 materials with a thickness of 16 mm. The test was carried out with a 70°, 4 MHz transverse wave head.

**Figure 15 materials-17-04216-f015:**
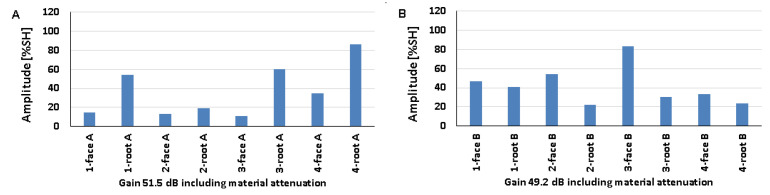
The change in signal amplitude depending on the position of the natural discontinuity for (**A**) 316L and (**B**) 13CrMo4-5 materials with a thickness of 16 mm. The test was carried out with a 60°, 4 MHz transverse wave head.

**Figure 16 materials-17-04216-f016:**
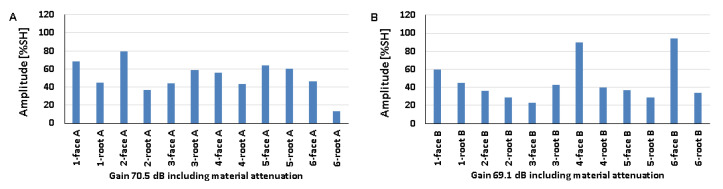
The change in signal amplitude depending on the position of the reference reflectors for (**A**) 13CrMo4-5 and (**B**) 316L materials with a thickness of 8 mm. The test was carried out with a 70°, 4 MHz longitudinal wave head.

**Figure 17 materials-17-04216-f017:**
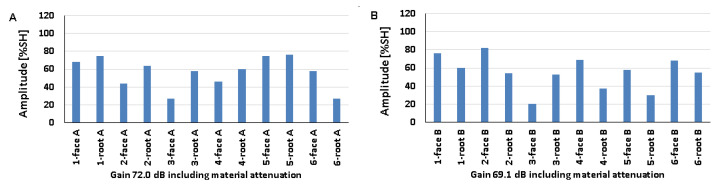
The change in signal amplitude depending on the position of the reference reflectors for (**A**) 13CrMo4-5 and (**B**) 316L materials with a thickness of 12 mm. The test was carried out with a 70°, 4 MHz longitudinal wave head.

**Figure 18 materials-17-04216-f018:**
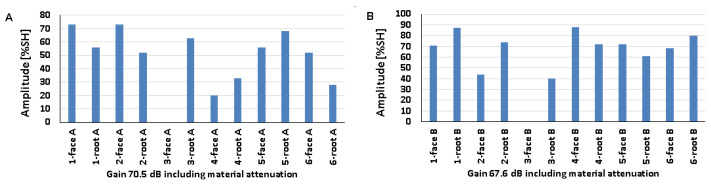
The change in signal amplitude depending on the position of the reference reflectors for (**A**) 13CrMo4-5 and (**B**) 316L materials with a thickness of 16 mm. The test was carried out with a 70°, 4 MHz longitudinal wave head.

**Figure 19 materials-17-04216-f019:**
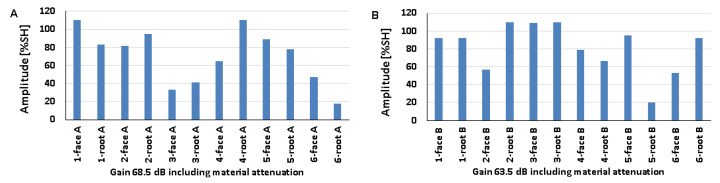
The change in signal amplitude depending on the position of the reference reflectors for (**A**) 13CrMo4-5 and (**B**) 316L materials with a thickness of 16 mm. The test was carried out with a 60°, 4 MHz longitudinal wave head.

**Figure 20 materials-17-04216-f020:**
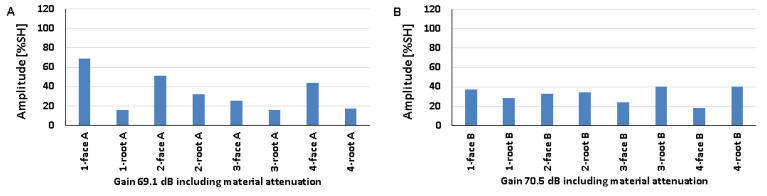
The change in signal amplitude depending on the position of the natural discontinuity for (**A**) 316L and (**B**) 13CrMo4-5 materials with a thickness of 8 mm. The test was carried out with a 70°, 4 MHz longitudinal wave head.

**Figure 21 materials-17-04216-f021:**
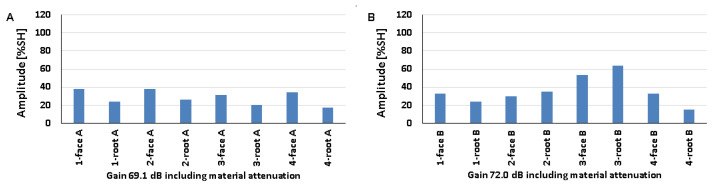
The change in signal amplitude depending on the position of the natural discontinuity for (**A**) 316L and (**B**) 13CrMo4-5 materials with a thickness of 12 mm. The test was carried out with a 70°, 4 MHz longitudinal wave head.

**Figure 22 materials-17-04216-f022:**
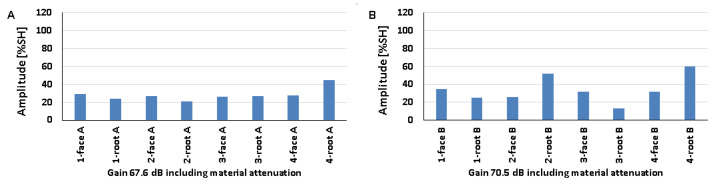
The change in signal amplitude depending on the position of the natural discontinuity for (**A**) 316L and (**B**) 13CrMo4-5 materials with a thickness of 16 mm. The test was carried out with a 70°, 4 MHz longitudinal wave head.

**Figure 23 materials-17-04216-f023:**
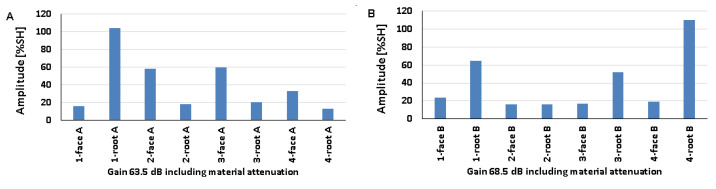
The change in signal amplitude depending on the position of the natural discontinuity for (**A**) 316L and (**B**) 13CrMo4-5 materials with a thickness of 16 mm. The test was carried out with a 60°, 4 MHz longitudinal wave head.

**Figure 24 materials-17-04216-f024:**
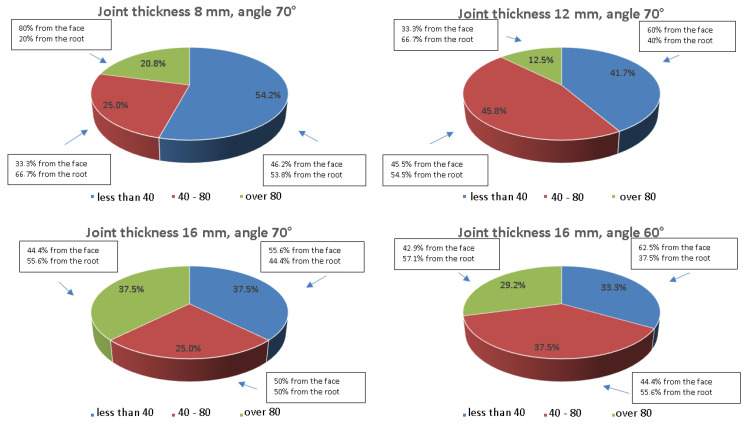
Percentage summary of the results when dividing into three ranges the values of the amplitudes obtained during the registration of the signal coming from the reference holes. Measurement was performed using a transverse wave head. In each interval, the percentage of how many indications were recorded from the boundary area and how many from the face was determined.

**Figure 25 materials-17-04216-f025:**
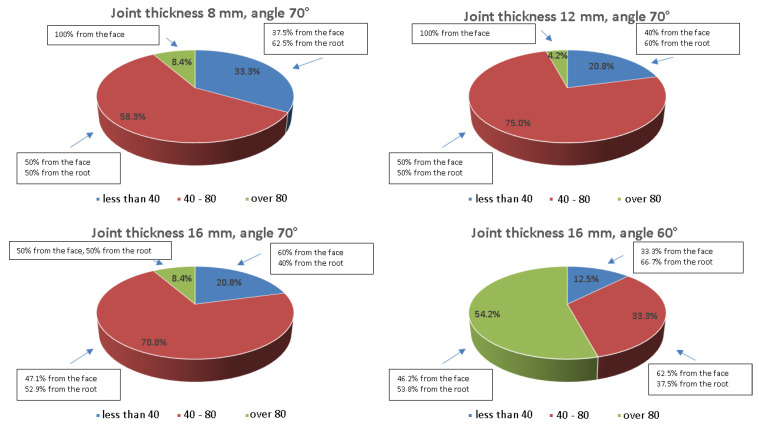
Percentage summary of the results for the diverse joints when divided into three ranges of amplitude values obtained when recording the signal coming from the reference holes. Measurement was performed with a longitudinal wave head. In each interval, the percentage of how many indications were recorded from the boundary area and how many from the face was determined.

**Figure 26 materials-17-04216-f026:**
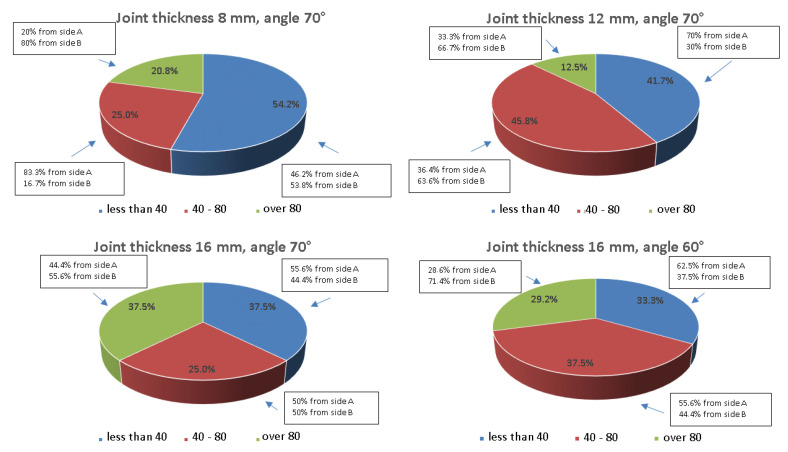
Percentage summary of the results for the diverse joints when divided into three ranges of amplitude values obtained during the recording of the signal coming from the reference holes. Measurement was performed with a transverse wave head. In each interval, the percentage of how many readings were recorded from side A (ferritic steel) and side B (austenitic steel) was determined.

**Figure 27 materials-17-04216-f027:**
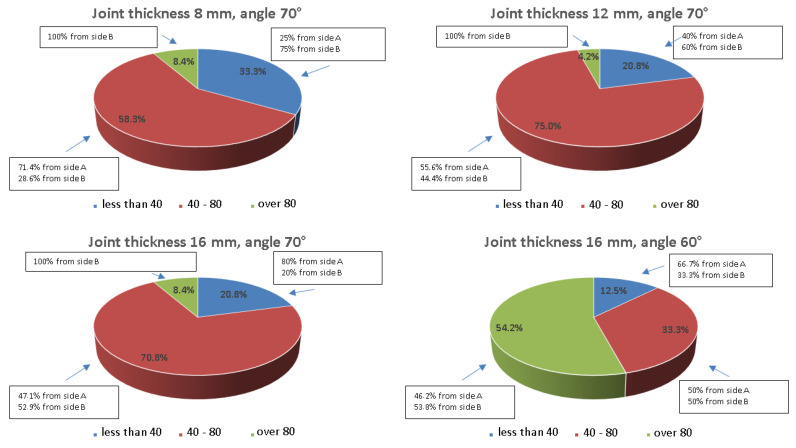
Percentage summary of the results for diverse joints when divided into three ranges of amplitude values obtained when recording the signal coming from the reference holes. Measurement was performed using a longitudinal wave head. In each interval, the percentage of how many readings were recorded from side A (ferritic steel) and side B (austenitic steel) was determined.

**Table 1 materials-17-04216-t001:** Indicative chemical composition of 13CrMo4-5 steel.

C, %	Cr, %	Ni, %	Mn, %	Mo, %	Si, %
0.16	0.8	0.03	0.475	0.471	0.213

**Table 2 materials-17-04216-t002:** Indicative chemical composition of X2CrNiMo17-12-2 (316L) steel.

C, %	Cr, %	Ni, %	Mn, %	Mo, %	N, %
≤0.03	17.5	11.5	≤2	2.3	≤0.11

**Table 3 materials-17-04216-t003:** Indicative chemical composition of LNM 309LSi wire.

C, %	Cr, %	Ni, %	Mn, %	Mo, %	Si, %
0.02	23.3	13.8	1.8	0.14	0.8

**Table 4 materials-17-04216-t004:** Selected physical properties of LNM 309LSi wire.

Parameter	Unit	Value
Average coefficient of thermal expansion at 20 ÷ 200 °C	1/K	16.5 × $10^{-6}$
Average coefficient of thermal expansion at 20 ÷ 400 °C	1/K	17.5 × $10^{-6}$
Thermal conductivity at 20 °C	W/(m K)	15
Specific heat capacity at 20 °C	J/(kg K)	500
Specific electrical resistance at 20 °C	Ω mm^2^/m	0.75
Density at 20 °C	kg/dm^3^	8.0
Modulus of elasticity at 20 °C	MPa	200

www.lincolnelectric.com/en/Products/lincolner309si309lsi_gtaw (accessed on 28 June 2024)

**Table 5 materials-17-04216-t005:** Summary of results of baseline and applied reinforcement for 13CrMo4-5–316L steel using transverse ultrasound waves.

Material Thickness t, mm	Angle, °	Base Gain Value, dB	Base Gain Value, dB	Used Gain Value, dB	Used Gain Value, dB
		**13CrMo4-5**	**316L**	**13CrMo4-5**	**316L**
8	70	47.1	44.1	52.0	51.0
12	70	47.1	44.1	52.7	50.8
16	70	47.1	44.1	54.5	51.1
16	60	39.7	40.5	49.2	51.5

**Table 6 materials-17-04216-t006:** Summary of results of baseline and applied reinforcement for 16 mm thick 13CrMo4-5–316L steel using longitudinal ultrasound waves.

Angle, °	Base Gain Value, dB	Base Gain Value, dB	Used Gain Value, dB	Used Gain Value, dB
	**13CrMo4-5**	**316L**	**13CrMo4-5**	**316L**
70	69.1	65.6	70.5	67.6
60	67.0	61.6	68.5	63.5

**Table 7 materials-17-04216-t007:** The change in the decibel gain level ($\Delta H_{u}$) with respect to the DAC comparison line for 316L material considering the path of the ultrasonic wave(s) for artificial discontinuities. The tests were carried out with a transverse ultrasound wave head.

Measu-	8 mm	8 mm	12 mm	12 mm	16 mm	16 mm	16 mm	16 mm
**rement**	**70°**	**70°**	**70°**	**70°**	**70°**	**70°**	**60°**	**60°**
**Point ***	** $\Delta H_{u}$ **	**s**	** $\Delta H_{u}$ **	**s**	** $\Delta H_{u}$ **	**s**	** $\Delta H_{u}$ **	**s**
	**dB**	**mm**	**dB**	**mm**	**dB**	**mm**	**dB**	**mm**
1-F-A	0.2	33.8	−1.8	22.2	1.4	24.6	6.2	45.9
1-F-B	3.4	30.2	3.1	44.4	1.0	24.3	7.8	42.0
1-R-A	−6.0	14.9	0.0	18.6	2.2	25.4	−0.2	15.0
1-R-B	−4.5	44.7	6.0	18.2	5.1	23.0	4.3	15.0
2-F-A	−5.3	32.7	0.1	49.7	−0.6	23.8	−1.8	16.0
2-F-B	−4.0	32.4	3.3	42.0	7.6	45.9	0.8	74.6
2-R-A	1.3	31.1	−9.5	54.8	1.6	24.4	4.3	14.8
2-R-B	−2.7	49.3	2.7	40.6	1.2	23.8	4.0	44.0
3-F-A	−5.8	38.8	−0.2	61.8	3.2	91.8	3.6	58.6
3-F-B	−6.4	32.4	4.2	49.8	4.6	74.1	9.3	56.1
3-R-A	−4.7	62.2	−0.9	28.9	1.8	43.0	−0.5	26.3
3-R-B	−4.9	52.4	−1.2	24.0	1.6	39.8	6.9	26.5
4-F-A	−7.5	36.2	−0.8	45.9	2.9	75.6	0.5	78.1
4-F-B	4.5	27.7	−9.1	43.8	3.6	23.0	−2.5	16.1
4-R-A	−10.0	18.7	0.0	24.5	2.4	26.1	−15.3	14.3
4-R-B	3.2	27.8	−14.5	43.2	2.2	22.1	4.2	13.9
5-F-A	6.0	17.0	−4.5	24.1	−0.1	52.7	5.2	19.7
5-F-B	−3.5	64.1	6.8	63.4	−9.4	31.9	−2.2	34.5
5-R-A	0.6	36.9	1.6	58.7	4.3	71.9	4.1	52.9
5-R-B	−2.1	47.4	−4.9	58.7	2.3	66.0	−5.7	60.2
6-F-A	−4.1	28.4	−6.4	29.8	0.3	32.7	−8.3	40.8
6-F-B	4.0	25.6	0.5	21.9	1.9	32.6	7.5	20.6
6-R-A	−3.5	44.8	−1.0	65.4	1.7	81.8	0.5	55.1
6-R-B	1.6	29.6	5.4	48.1	2.0	65.8	9.3	62.8

* 1–6—locations of measurement points according to Figure 3; F/R—face/ridge of the weld; A/B—sides of the welded joint according to Figure 3.

**Table 8 materials-17-04216-t008:** The change in the decibel gain level ($\Delta H_{u}$) with respect to the DAC comparison line for 316L material considering the path for the ultrasonic wave(s) for natural discontinuities. The tests were carried out with a transverse ultrasound wave head.

Measu-	8 mm	8 mm	12 mm	12 mm	16 mm	16 mm	16 mm	16 mm
**rement**	**70°**	**70°**	**70°**	**70°**	**70°**	**70°**	**60°**	**60°**
**Point ***	** $\Delta H_{u}$ **	**s**	** $\Delta H_{u}$ **	**s**	** $\Delta H_{u}$ **	**s**	** $\Delta H_{u}$ **	**s**
	**dB**	**mm**	**dB**	**mm**	**dB**	**mm**	**dB**	**mm**
1-F-A	−7.8	40.2	−7.3	37.9	−2.6	59.0	−4.7	47.3
1-F-B	0.0	0.0	−18.1	25.5	0.0	0.0	2.8	55.5
1-R-A	−13.5	30.8	−5.4	60.6	−9.4	24.3	−0.4	24.8
1-R-B	−15	33.9	−20.0	22.2	−8.7	25.3	−3.3	26.1
2-F-A	−1.8	30.3	−5.9	59.3	−10.2	48.0	−7.3	38.8
2-F-B	−12.2	55.5	−18.9	19.7	−24.8	24.8	2.4	48.8
2-R-A	−3.0	49.5	−8.6	31.6	−12	17.9	−11.7	49.6
2-R-B	−15.4	57.7	−14.9	29.5	−12.2	24.2	−3.7	16.8
3-F-A	−14.8	40.4	−6.1	45.7	−3.2	47.7	−11.2	37.6
3-F-B	−11.5	54.1	−18.9	18.7	0.0	0.0	5.7	44.5
3-R-A	−4.3	57.8	−12.0	34.4	−12.5	53.3	−0.1	20.9
3-R-B	0.0	0.0	−12.9	43.3	6.8	35.9	−6.8	23.2
4-F-A	−11.8	41.6	−4.7	59.8	−5.3	54.3	2.1	39.8
4-F-B	−12	30.1	−14.8	80.5	−4.4	71.7	−1.8	50.1
4-R-A	−12	13.3	−7.0	56.7	−8.8	59.4	1.9	34.8
4-R-B	−13	10.0	−20.6	44.0	−3.1	29.9	−5.8	18.7

* 1–4—location of measurement points according to Figure 3; F/R—face/ridge of the weld; A/B—sides of the welded joint according to Figure 3.

**Table 9 materials-17-04216-t009:** The change in the decibel gain level ($\Delta H_{u}$) with respect to the DAC comparison line for 316L material considering the path for the ultrasonic wave(s) for artificial discontinuities. The tests were carried out with a longitudinal ultrasound wave head.

Measu-	8 mm	8 mm	12 mm	12 mm	16 mm	16 mm	16 mm	16 mm
**rement**	**70°**	**70°**	**70°**	**70°**	**70°**	**70°**	**60°**	**60°**
**Point ***	** $\Delta H_{u}$ **	**s**	** $\Delta H_{u}$ **	**s**	** $\Delta H_{u}$ **	**s**	** $\Delta H_{u}$ **	**s**
	**dB**	**mm**	**dB**	**mm**	**dB**	**mm**	**dB**	**mm**
1-F-A	0.4	25.4	0.3	25.1	1.6	28.3	5.1	21.8
1-F-B	0.0	23.6	1.7	24.7	1.6	25.5	4.8	32.6
1-R-A	−3.9	22.3	1.1	24.6	−0.2	30.3	8.4	21.4
1-R-B	−3.9	19.6	−1.0	20.6	3.0	23.6	3.6	21.8
2-F-A	−3.9	19.9	−2.3	30.4	0.5	22.3	2.9	26.0
2-F-B	−3.5	30.0	1.6	20.1	−0.7	37.1	0.0	25.9
2-R-A	−4.9	20.8	0.3	27.9	−1.4	28.7	4.2	25.9
2-R-B	−5.1	20.9	−2.6	16.7	2.8	30.4	5.5	16.4
3-F-A	−3.6	24.7	−8.0	24.2	0.0	0.0	−4.7	31.0
3-F-B	−9.2	21.5	−8.8	27.3	0.0	0.0	4.3	19.7
3-R-A	−1.1	24.3	−2.0	23.7	2.6	45.3	−3.3	24.3
3-R-B	−3.1	24.5	−0.9	26.1	−1.4	43.4	5.3	15.3
4-F-A	−1.8	22.9	−4.3	19.3	0.0	34.9	1.5	32.9
4-F-B	2.5	19.2	2.1	29.8	3.0	23.8	3.5	32.6
4-R-A	−3.0	27.8	−2.2	17.6	−4.3	33.4	9.6	33.4
4-R-B	−5.6	15.5	−4.6	22.9	1.7	25.5	−4.7	26.5
5-F-A	−0.9	21.5	2.2	29.5	1.3	39.8	9.8	28.3
5-F-B	−4.0	26.0	1.8	33.8	3.0	33.2	3.9	22.6
5-R-A	−1.9	19.6	−0.4	16.3	0.0	23.6	0.3	12.7
5-R-B	−7.1	23.6	−5.9	26.5	0.1	23.0	−17.3	14.9
6-F-A	−2.3	29.8	0.7	33.4	0.0	35.8	4.6	34.1
6-F-B	2.4	17.7	0.6	24.7	3.4	41.6	2.0	35.2
6-R-A	−15.3	21.9	−2.4	29.4	−7.5	23.5	−10.3	25.7
6-R-B	−7.3	13.2	−3.2	13.1	−0.1	12.7	−4.3	11.6

* 1–6—location of measurement points according to Figure 3; F/R—face/ridge of the weld; A/B—sides of the welded joint according to Figure 3.

**Table 10 materials-17-04216-t010:** The change in the decibel gain level ($\Delta H_{u}$) with respect to the DAC comparison line for 316L material considering the path for the ultrasonic wave(s) for natural discontinuities. The tests were carried out with a longitudinal ultrasound wave head.

Measu-	8 mm	8 mm	12 mm	12 mm	16 mm	16 mm	16 mm	16 mm
**rement**	**70°**	**70°**	**70°**	**70°**	**70°**	**70°**	**60°**	**60°**
**Point ***	** $\Delta H_{u}$ **	**s**	** $\Delta H_{u}$ **	**s**	** $\Delta H_{u}$ **	**s**	** $\Delta H_{u}$ **	**s**
	**dB**	**mm**	**dB**	**mm**	**dB**	**mm**	**dB**	**mm**
1-F-A	1.6	26.8	−1.6	40.3	−4.3	38.9	−10.7	26.5
1-F-B	−8.8	22.1	−4.4	32.1	−3.1	38.0	−8.9	18.7
1-R-A	−11.8	23.2	−6.1	37.2	−1.8	25.6	5.5	32.9
1-R-B	−6.0	30.3	−7.4	29.6	−6.5	36.3	0.4	21.6
2-F-A	0.2	30.1	−2.3	35.9	−5.2	37.7	0.3	33.5
2-F-B	−5.7	24.9	−5.4	30.8	−5.3	40.1	−11.8	20.7
2-R-A	−5.6	23.4	−5.2	38.7	−7.1	38.6	−11.4	17.0
2-R-B	−5.2	26.7	−4.9	29.0	0.3	36.8	−11.8	16.0
3-F-A	−7.1	28.0	−3.7	42.5	−5.5	38.0	0.6	33.8
3-F-B	−7.3	32.9	−0.3	32.6	−3.8	38.6	−10.9	26.0
3-R-A	7.4	25.7	−9.0	28.5	−8.3	20.7	−8.9	28.2
3-R-B	−3.6	28.4	0.9	30.4	−15.1	23.3	−0.9	27.9
4-F-A	−1.5	32.2	−2.6	40.9	−4.8	37.7	−4.7	33.0
4-F-B	−12.0	20.8	−5.0	30.4	−4.0	36.5	−9.5	28.1
4-R-A	−10.7	27.3	−9.5	34.0	−3.3	20.6	−15.7	16.2
4-R-B	−4.0	27.0	−11.5	26.9	−1.3	23.5	7.4	33.0

* 1–4—location of measurement points according to Figure 3; F/R—face/ridge of the weld; A/B—sides of the welded joint according to Figure 3.

**Table 11 materials-17-04216-t011:** Summary of measurement points with consideration of their detectability (●—poorly detectable, amplitude below 40% SH; ●●—well detectable, amplitude between 40 and 80% SH; ●●●—very well detectable, amplitude above 80% SH). Data are presented for all tested joint thicknesses and both angles using ultrasonic transverse (T) and longitudinal (L) wave heads.

Measurement	8, mm	12, mm	16, mm	16, mm	8, mm	12, mm	16, mm	16, mm	∑	∑	∑
point	70°, T	70°, T	70°, T	60°, T	70°, L	70°, L	70°, L	60°, L	●	●●	●●●
1-weld face A	●●	●●	●●●	●●	●●	●●	●●	●●●	0	6	2
1-weld face B	●●●	●●	●●●	●●●	●●	●●	●●	●●●	0	4	4
1-weld root A	●●	●●	●●●	●●	●●	●●	●●	●●●	0	6	2
1-weld root B	●	●●●	●●	●●●	●●	●●	●●●	●●●	1	3	4
2-weld face A	●	●	●●	●●	●●	●●	●●	●●●	2	5	1
2-weld face B	●	●	●●●	●	●	●●	●●	●●	4	3	1
2-weld root A	●●	●●●	●●●	●●●	●	●●	●●	●●●	1	3	4
2-weld root B	●	●	●●●	●●	●	●	●●	●●●	4	2	2
3-weld face A	●	●	●	●	●●	●	●	●	7	1	0
3-weld face B	●	●●	●	●●●	●	●	●	●●●	5	1	2
3-weld root A	●	●●	●●	●●	●●	●●	●●	●●	1	7	0
3-weld root B	●	●●	●●	●●●	●●	●●	●●	●●●	1	5	2
4-weld face A	●	●	●	●	●●	●●	●	●●	5	3	0
4-weld face B	●●●	●	●●●	●●	●●●	●●●	●●●	●●	1	2	5
4-weld root A	●	●	●●●	●	●●	●●	●	●●●	4	2	2
4-weld root B	●●●	●●	●●●	●●●	●●	●●	●●	●●	0	5	3
5-weld face A	●●●	●●	●	●●●	●●	●●	●●	●●●	1	4	3
5-weld face B	●	●●	●	●	●	●●	●●	●●●	4	3	1
5-weld root A	●●	●	●	●●	●●	●●	●●	●●	2	6	0
5-weld root B	●	●●	●	●	●	●	●●	●	6	2	0
6-weld face A	●●	●	●●	●	●●	●●	●●	●●	2	6	0
6-weld face B	●●●	●●●	●●	●●	●●●	●●	●●	●●	0	5	3
6-weld root A	●	●	●	●	●	●	●	●	8	0	0
6-weld root B	●●	●●	●	●●	●	●●	●●	●●●	2	5	1
∑ ●	13	10	9	8	8	5	5	3	-	-	-
∑ ●●	6	11	6	9	14	18	17	8	-	-	-
∑ ●●●	5	3	9	7	2	1	2	13	-	-	-

## Data Availability

The original contributions presented in the study are included in the article, further inquiries can be directed to the corresponding authors.

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
