# Peer review of "Analysis of the Suitability of Ultrasonic Testing for Verification of Nonuniform Welded Joints of Austenitic–Ferritic Sheets"

_materials, 2024, doi:10.3390/ma17174216_

Round 1
Reviewer 1 Report
Comments and Suggestions for Authors
The manuscript is very poorly written. It is apparent that the authors have not proof-read their manuscript before submission. I have provided my detailed comments in the attached pdf.

The quality of English is poor. Sometimes it was difficult to understand what the authors wanted to convey. It is recommended that the authors use the English editing services to significantly improve the quality of English in their manuscript.
Author Response
Dear Reviewer,
please see the attachment.
Yours faithfully
Grzegorz Peruń

Reviewer 2 Report
Comments and Suggestions for Authors
The authors presented very a detailed technical report demostrating the results of the experimental studies provided by the authors. I cannot report sufficient scientific novelty in the paper, though the authors provided a large number of measurements using Olympus EPOCH 650 ultrasonic defectoscope with various heads and various speciments.
Nevertheless, the paper might be recommended for publication if the authors will be able to formulate more comprehensive conclusion, which generalize their studies and also reply to the following comments.
- Why beam introduction head at an angle of 70° and 60° was chosen?
- It seems that welded joints shown in Figure 4 have dissimilar angle between two welded joint. Is it right? If so, how it might affect the obtained results.
- The reviewer has not realized why beam introduction head at an angle of 60° was used for 16mm thick specimen.
- Prior to depicting Figures 22-27, it would be benefitial to provide explicit formulae for the calculated percentage.
Author Response

(The authors gave the same response as above.)

Round 2
Reviewer 1 Report
Comments and Suggestions for Authors
The authors have addressed all of my previous comments in sufficient detail. I have no further feedback.
Comments on the Quality of English LanguageThe quality of English is a lot better than the original manuscript version. However, a few more grammatical errors still present throughout the manuscript which the MDPI's English Editor will be able to correct.
Reviewer 2 Report
Comments and Suggestions for Authors
Accept.